# **Review article: Kinematic models of the interplanetary magnetic field**

## Christoph Lhotka<sup>1</sup> and Yasuhito Narita<sup>1</sup>

<sup>1</sup>Space Research Institute, Austrian Academy of Sciences, Schmiedlstr. 6, A-8042 Graz, Austria **Correspondence:** C. Lhotka (christoph.lhotka@oeaw.ac.at)

Abstract. Current knowledge on the description of the interplanetary magnetic field is reviewed with an emphasis on the kinematic approach as well as the analytic expression. Starting with the Parker spiral field approach, further effects are incorporated into this fundamental magnetic field model, including the latitudinal dependence, the poleward component, the solar cycle dependence, and the polarity and tilt angle of the solar magnetic axis. Further extensions are discussed in view of the magnetohydrodynamic treatment, the turbulence effect, the pickup ions, and the stellar wind models. The models of the

5 the magnetohydrodynamic treatment, the turbulence effect, the pickup ions, and the stellar wind models. The models of the interplanetary magnetic field serve as a useful tool for theoretical studies, in particular on the problems of plasma turbulence evolution, charged dust motions, and cosmic ray modulation in the heliosphere.

### 1 Introduction

The interplanetary magnetic field (IMF) is a spatially extended magnetic field of the Sun, and forms together with the plasma

- flow from the Sun (referred to as the solar wind) a spatial domain of the *heliosphere*<sup>1</sup> around the Sun surrounded by the local interstellar cloud. Starting with the first direct measurements in 1960's (Ness et al., 1964; Ness and Wilcox, 1964; Wilcox and Ness, 1965; Wilcox, 1968), the IMF is becoming increasingly more accessible in various places in situ in the solar system, e.g., the inner heliosphere (closer than the Earth orbit from the Sun) was covered by the Helios mission (Porsche, 1981), see monograph by Schwenn and Marsch (1990, 1991), the outer heliosphere (beyond the Earth orbit) by Voyager (Stone, 1977;
- Kohlhase and Penzo, 1977; Stone, 1983), and the high-latitude region by the Ulysses mission (Wenzel and Smith, 1991; Wenzel et al., 1992).

In the lowest-order picture, IMF has an Archimedian spiral structure, also referred to as the Parker spiral after Parker (1958), imposed by the solar wind expansion and the solar rotation, and exhibits spatial variation (e.g., sectors with the opposite directions of the radial component of the magnetic field, latitude dependence) and time variation (e.g., solar cycle dependence).

Typical values of the IMF magnitude (in the sense of the mean field)  $B_0$  turn out to be of the order of 3–4 nT at the Earth orbit (1 astronomical unit, hereafter au). Long-term measurements of the IMF by the Ulysses spacecraft show that the field magnitude of about 3–4 nT is typical not only in the solar ecliptic plane but also in the high-latitude regions (Forsyth et al., 1996). Of course, irregular or transient phenomena (such as coronal mass ejections or co-rotating interaction regions)

<sup>&</sup>lt;sup>1</sup>IMF is also referred to as the heliospheric magnetic field.

cause local, large-amplitude deviations from the mean field. Recent study by Henry et al. (2017) indicates that the IMF (at the Earth orbit) can be regarded as the Parker spiral type when the IMF is sufficiently inclined to the Earth orbital plane, either (1)  $B_x > 0.4B$  and  $B_y < -0.4B$  or (2)  $B_x < -0.4B$  and  $B_y > 0.4B$ , where  $B_x$  is the sunward component of the magnetic field (GSE-X direction),  $B_y$  is the dawn-to-dusk component of the field (GSE-Y direction), and B is the magnetic field

5 magnitude. The IMF can be more radial and of the Ortho-Parker spiral type (valid under  $|B_x| > 0.4B_t$ , where  $B_t$  denotes the transverse component of the magnetic field to the radial direction from the Sun,  $B_t = \sqrt{B_y^2 + B_z^2}$ ) or oriented more northward or southward  $|B_z| > 0.5B_t$ .

Model construction of the IMF has immediate applications in the following plasma physical or astrophysical problems:

- 1. Solar wind turbulence.
- Plasma and magnetic field in interplanetary space develop into turbulence. Early in situ measurements in 1960's have already shown that the frequency spectrum of the fluctuation of the IMF is a power-law over a wide range of frequencies (typically in the mHz regime) (Coleman, 1968), and the spectral index is close to -5/3 (Matthaeus et al., 1982; Tu and Marsch, 1995), known as the inertial-range spectrum of fluid turbulence. Properties of solar wind turbulence are extensively studied using in situ spacecraft such as Helios, Voyager, Ulysses, and the observational properties are documented in reviews by, e.g., Tu and Marsch (1995), Petrosyan et al. (2010), and Bruno and Carbone (2013). Solar wind is the only accessible natural laboratory of turbulence in collisionless plasmas, relevant to astrophysical applications to interstellar turbulence. Knowledge on the IMF structure is an important ingredient in turbulence modeling. In particular, the large-scale inhomogeneity or velocity shear are the driver of turbulence when the solar wind plasma evolves into turbulence. For example, the mean-field models of turbulence explicitly need the large-scale structure as an input (Yokoi and Hamba, 2007; Yokoi, 2011).
  - 2. Charged dust motion.

Dust grains in interplanetary space have typically a length scale of nanometers to micrometers, and are electrically charged by various processes, e.g., sticking of the ambient electrons onto the dust surface (which makes the dust charge state negative) or photo-electrons (which makes the charge state positive) (Shukla, 2001; Mann et al., 2014). Unlike the electrons or ions in the plasma, the charged dust grains undergo not only the gravitation attraction by the Sun and the planets and the Poynting-Robertson effect but also the electromagnetic interaction (Coulomb and Lorentz force). Combination of these forces results, e.g., in a long-time tilt of the orbital plane (on the time scale of 10 to 100 years), e.g., perihelion or apohelion shift from the solar ecliptic plane to the high-latitude region. Knowledge on the IMF structure is important because the orbital motion and the orbit drift can be tracked, either in a static IMF structure or in a time-evolving IMF structure (Grün et al., 1994; Mann et al., 2007, 2014; Czechowski and Mann, 2010; Lhotka et al., 2016).

30

25

3. Cosmic ray modulation.

Cosmic ray consists mostly (more than 90%) of protons. The spectrum of the cosmic ray is well characterized by a power-law as a function of the particle energy (kinetic energy, strictly speaking) with a peak at about 1 GeV and a

slope of about -2.7. The number flux of the cosmic ray can be measured by the neutron monitors, and is known to be anti-correlated to the sunspot number variations with a period of about 22 years (cosmic ray modulation). The cosmic ray transport in the heliosphere is modeled by the convection-diffusion equation system, which can be treated both in a kinetic way based on the Boltzmann transport theory (Parker, 1965) and in a fluid-physical way using the continuity equation with the convection and diffusion terms (Duldig, 2001). See also the recent review by Potgieter (2013). The knowledge of IMF is important because the cosmic ray exhibits charged particles undergo drift motions in a curved, inhomogeneous magnetic field (i.e., curvature drift and grad-B drift), as pointed out by, e.g., Isenberg and Jokipii (1979). In fact, the 22-year variation of the cosmic ray modulation (as measured by the neutron monitors on the Earth ground) can be explained and theoretically reconstructed by including the IMF structure (Kóta and Jokipii, 2001a; Burger et al., 2008; Miyahara et al., 2010).

10

15

5

Here we review various models of the IMF with an emphasis on the hydrodynamic approach and the analytic expression. This review is intended to complement a more comprehensive review by Owens and Forsyth (2013). We limit our review to the kinematic approach in the sense that the magnetic fields behave passively and are frozen-in into the given plasma flow. The review is organized in a concise way by primarily taking the kinematic approach. There is an increasing amount of literatures and studies about the IMF and the modeling approach is becoming diverse, e.g., hydrodynamic, hydromagnetic, and kinetic. We point out, however, that even in the simple kinematic approach, the IMF models are still illustrative and have various

applications as introduced above.

We also limit our review to the analytic expression as much as possible. Analytic expression of the magnetic fields is a useful tool in space science, and has been constructed for various plasma domains or plasma phenomena in the solar system other than the solar wind: solar corona (Banaszkiewicz et al., 1998), coronal mass ejection (CME) (Isavnin, 2017), Earth's

20 other than the solar wind: solar corona (Banaszkiewicz et al., 1998), coronal mass ejection (CME) (Isavnin, 2017), Earth's magnetosphere (Katsiaria and Psillakis, 1987; Tsyganenko, 1990, 1995; Tsyganenko and Sitnov, 2007), and local interstellar medium surrounding the heliosphere (Röken, 2015). One can of course numerically solve the governing equations to reproduce the magnetic field and its dynamics more realistically, but the numerical treatment is not the scope of this review.

The advantage of the analytic or semi-analytic expression is that one can implement the magnetic field models by themselves for the theoretical studies of the solar system plasma phenomena. Verification of the magnetic field models is possible using the existing in situ spacecraft data from, e.g., the Helios, Voyager, and Ulysses missions as well as the upcoming measurements in interplanetary space by Parker Solar Probe (Fox et al., 2016), BepiColombo's cruise in interplanetary space (Benkhoff et al., 2010), and Solar Orbiter (Müller et al., 2013).

#### 2 Kinematic approach

30 We focus on the kinematic approach such that the flow pattern is given as an external field of a model field. The magnetic field is passive in the sense of the frozen-in field into the plasma. The reaction of the magnetic field onto the plasma motion (such as the Lorentz force acting on the plasma bulk flow) is not considered here.

#### 2.1 Parker model

#### 2.1.1 Thermally-driven wind

In this section we review the formulation of the original Parker spiral model of the interplanetary magnetic field. As suggested by Biermann (1951, 1957) the solar gas outflows into interplanetary space. The existence of the radial outflow of the solar

- 5 gaseous material, nowadays known as the solar wind, and the spiral structure of the IMF associated with the solar rotation were predicted by Parker (1958) before the confirmation by in situ spacecraft measurements. It is worth while to note that the spiral structure in interplanetary space was also indicated in the comet tail study by Alfvén (1957) as a beam extending away from the Sun. The solar wind is mainly composed of protons, electrons, and helium alpha particles (there are, in addition, heavier ions from the Sun and pickup ions from the local interstellar medium), and streams radially away from the Sun far beyond the
- orbits of the planets over distances of about 100 au. The solar wind first encounters the termination shock located before the heliopause, a boundary layer between the solar plasma and the local interstellar medium at a distance of about 110–160 au. At the Earth orbit distance (1 au), the solar wind velocity typically ranges between 300 km s<sup>-1</sup> (referred to as the slow solar wind) to 700 km s<sup>-1</sup> (the fast solar wind). During the coronal mass ejection events, the solar wind speed can reach about 1400 km s<sup>-1</sup>.
- The Parker model treats the solar wind as a one-dimensional (in the radial direction), steady-state, iso-thermal thermallydriven stream. Basic equations are the continuity equation,

$$\frac{\mathrm{d}}{\mathrm{d}r}\left(\rho U_r r^2\right) = 0,\tag{1}$$

the momentum balance,

$$U_r \frac{\mathrm{d}U_r}{\mathrm{d}r} + \frac{1}{\rho} \frac{\mathrm{d}p}{\mathrm{d}r} + \frac{GM_{\odot}}{r^2} = 0,\tag{2}$$

and the adiabatic law or the equation of state,

$$p = \rho c_{\rm s}^2. \tag{3}$$

Here  $\rho$  denotes the mass density,  $U_r$  the radial component of the flow velocity, r the distance from the Sun, p the gas pressure, G the gravitational constant,  $M_{\odot}$  the solar mass, and  $c_s$  the sound speed. Note that the sound speed is considered constant due to the assumption of the iso-thermal medium. Equations (1)–(3) can be reduced into the following form,

$$U_r \frac{\mathrm{d}U_r}{\mathrm{d}r} = \left(\frac{2c_{\mathrm{s}}^2}{r} - \frac{GM}{r^2}\right) \left(1 - \frac{c_{\mathrm{s}}^2}{U_r^2}\right)^{-1}.$$
 (4)

One sees immediately that Eq. (4) has a singularity at which  $U_r = c_s$  is satisfied. The flow speed reaches the sound speed (called the critical point or the sonic point) at

$$r_{\rm c} = \frac{GM_{\odot}}{2c_{\rm s}^2}.$$
(5)

The critical point is located about 6 solar radii for a (coronal) temperature of 1 MK. Equation (4) exhibits difference types or classes of the flow velocity profile as a function of the distance from the Sun. Above all, a continuous flow acceleration over the sonic point meets the condition for the solar wind, i.e., acceleration in the subsonic domain ( $r < r_c$ ) and further acceleration in the supersonic domain ( $r > r_c$ ). See, e.g., Tajima and Shibata (2002) for a more detailed description about the Parker model. At a larger distance than the critical radius  $r_c$ , the flow velocity has an asymptotic form,

$$U_r \simeq 2c_{\rm s} \left( \ln \frac{r}{r_c} \right)^{1/2}.\tag{6}$$

A comparison between the approximation of  $U_r$  using (6) and a numerical solution of (4) is shown in Fig. 1. The solution shown in red and obtained for T = 1MK, perfectly agrees with the analytical solution shown in dashed black. The Parker model thus predicts that the solar corona expands radially outward at subsonic velocities close to the Sun (within the critical

10 radius), and the coronal gas is gradually accelerated to supersonic velocities further out. Hereafter we also use an expression of  $U_{sw}$  for the magnitude of the solar wind velocity. A more detailed analysis of the Parker model with the asymptotic solution of the flow velocity is presented by Summers (1978). A two-fluid model of the solar wind is presented by Summers (1982) as a hydrodynamic extension of the Parker model for the electron and the protons under the adiabatic law for each fluid type.

#### 2.1.2 Spiral magnetic field

5

15 Using the angular velocity of the Sun,  $\Omega_{\odot}$ , the radial, polar, and azimuthal components of the solar wind velocity is given in the HG (heliographic) frame of reference as follows,

$$U_r = U_{\rm sw} \tag{7}$$

$$U_{\theta} = 0 \tag{8}$$

$$U_{\phi} = -\Omega_{\odot} r \sin\theta. \tag{9}$$

20 A magnetic stream line satisfies the differential equation at a given polar angle  $\theta$ ,

$$\frac{1}{r\sin\theta}\frac{\mathrm{d}r}{\mathrm{d}\phi} \simeq \frac{U_r}{U_\phi} = -\frac{U}{\Omega_\odot r\sin\theta}.$$
(10)

We make use of a rough assumption that the flow speed is nearly constant over the critical radius beyond some distance  $r > r_c$ . The field-line equation (Eq. 10) has then the solution as

$$r - r_0 = -\frac{U_{\rm sw}}{\Omega_{\odot}} \left(\phi - \phi_0\right). \tag{11}$$

Here, the magnetic field line passes through the coordinate at  $(r_0, \theta, \phi_0)$ . The IMF is obtained from the divergence-free condition of the Maxwell equations,

$$\nabla \cdot \boldsymbol{B} = \boldsymbol{0}. \tag{12}$$

Figure 1. Radial solution of the solar wind  $U_r$  for different temperatures in mega-Kelvins (right frame ticks). Vertical lines indicate the position of the planets, the dark-shaded region covers the region of main belt asteroids of the solar system, where blue lines mark the position of mean motion resonances of asteroids with planet Jupiter.

That is, using the assumption of spherically symmetry, the IMF is expressed as

$$B_r = B_0 \left(\frac{r_0}{r}\right)^2 \tag{13}$$

$$B_{\theta} = 0 \tag{14}$$

$$B_{\phi} = -B_0 \frac{M_{\odot} r_0}{U_{\rm sw}} \frac{r_0}{r} \sin\theta, \tag{15}$$

5 where  $B_0$  is the radial component of the magnetic field at a reference radius  $r_0$ . The transformation into the stationary frame (HGI, heliographic inertial) yields the same expression of the magnetic field as Eqs. (13)–(15). Note that due to a Galilean transformation, the electric field has a convective contribution in the polar direction  $e_{\theta}$ ,

$$\boldsymbol{E} = -\boldsymbol{U} \times \boldsymbol{B} = -U_{\rm sw} B_{\boldsymbol{\theta}} \boldsymbol{e}_{\boldsymbol{\theta}}.\tag{16}$$

Realizations of the magnetic field lines in the Parker spiral model are shown for different (constant) solar wind speeds in Fig. 2.
 The angle between the the magnetic field line and Earth's orbit is about 45° for a typical solar wind speed of 400 km s<sup>-1</sup>, and increases (becomes more radial) at a higher flow speed. Note that when considering the magnetohydrodynamic (MHD) effect,