# Peer review of "Review article: Kinematic models of the interplanetary magnetic field"

_Annales Geophysicae, 2018_

## Referee Comment (RC1) · Anonymous Referee #1 · 11 Mar 2019

This draft reviews basic concepts of the large-scale heliospheric magnetic fields mainly from the kinematic framework. I have the following (mostly minor) comments:

1. Fig.2 The blue, red, and gray lines should be defined in the caption.

2.The caption of Fig.3 The authors should explain the line types and the numbers (24.47h, etc.: Rotational period) in the caption or in the panel.

3. Fig.4 and Section 2.1.4 In the section 2.1.4, it seems that the authors focuses on the situation, in which B\_{\theta}=0 (eq.20). But Figure 4 shows field lines with non-zero B\_{\theta}. Probably, the formulation of eqs.(22) - (25) in the same section takes into account B\_{\theta}. I think more explanations are necessary, which is friendly to readers.

4. eq.(49) The scaling should be  $r^{-1}$ , instead of r. (I think this is simply a typo.)

5. eq.(50) It is probably better to refer to old works (Alazraki & Couturier 1971; Belcher 1971), in addition to the recent works that are already cited in the present paper.

---

## Referee Comment (RC2) · Anonymous Referee #2 · 20 Mar 2019

The article is a very nice tutorial overview of the subject. The grammar and spelling need to be reviewed, an example: page 3, line 19 "useful took" presumably should be "useful tool". I will leave this for the editorial staff and authors to go through this instead of providing an incomplete list.

For the physics discussion, last section should really be expanded a little more to be a re view rather than a tutorial. I would like to see a little more discussion of the two dimensional treatment, turbulent diffusion, as well as pickup ion effects on pages 16 and 17. A comprehensive review should include a basic discussion of the models. Authors already have a lot of the references in there. Including the model equations and a basic discussion of how they incorporate the higher order effects would make this review a good one stop overview read. I would like to see an expanded section 3.

[Figure]

I had minor comments on figures and captions but the other referee has already discussed them in more detail than I was planning.

---

## Author Comment (AC1) · 4 Apr 2019

The comment was uploaded in the form of a supplement:
https://www.ann-geophys-discuss.net/angeo-2018-133/angeo-2018-133-AC1-supplement.pdf

---

## Author Comment (AC2) · 4 Apr 2019

**Reply to referee-2 comments**

Thank you very much for the positive evaluation. Here are reply comments.

• The article is a very nice tutorial overview of the subject. The grammar and spelling need to be reviewed, an example: page 3, line 19 "useful took" presumably should be "useful tool". I will leave this for the editorial staff and authors to go through this instead of providing an incomplete list.

**Reply**: Thank you for the positive evaluation and a careful check of the manuscript text.

- "useful took" was corrected into "useful tool" (page 4).
- We went through the spelling check and the sentence check to eliminate errors in English.
- For the physics discussion, last section should really be expanded a little more to be a review rather than a tutorial. I would like to see a little more discussion of the two dimensional treatment, turbulent diffusion, as well as pickup ion effects on pages 16 and 17. A comprehensive review should include a basic discussion of the models. Authors already have a lot of the references in there. Including the model equations and a basic discussion of how they incorporate the higher order effects would make this review a good one stop overview read. I would like to see an expanded section 3.

**Reply**: Agreed. We added the following text and explanations.

 A model including the latitude dependence (Lima et al., 2001) is added to section 2.2.1. (page 12).

"A model of latitudinal dependence of the magnetic field is constructed by employing the method of separation of variable for an axi-symmetric magnetohydrodynamic outflow (Lima et al., 2001). The radial and the azimuthal components of the magnetic field are proposed as

$$B_r = \frac{B_0}{r^2} \sqrt{1 + \mu \sin^{2\epsilon} \theta}$$
(41)

$$B_{\phi} = \lambda B_0 \frac{\sin^{\epsilon} \theta}{r} \left( \frac{\frac{r^2}{R_s^2} - 1}{1 - M_A^2} \right), \qquad (42)$$

where  $\epsilon$  is a free parameter,  $\mu$  is the ratio of the flow kinetic energy (or energy density, strictly speaking) in the equatorial region to that in the polar region, and  $\lambda$  is the ratio of azimuthal to radial velocity (and also magnetic field) at the base of the wind.  $R_s$  is the radius of the star or the Sun.  $M_A$  is the Alfvén Mach number of the flow. The polar component of the magnetic field is assumed to vanish due to the assumption of the axial symmetry around the rotation axis."  A model including the tilt angle and the solar cycle dependence (Burger et al., 2008) is added to section 2.2.4. (page 16 to page 18).

"A more refined magnetic field model is constructed by Burger et al. (2008), which offers an extension of the tilted heliospheric current sheet (with respect to the rotation axis) to the solar cycle dependence. The latitude-dependent magnetic field model is expressed as follows:

$$B_r = B_0 \left(\frac{r_0}{r}\right)^2 \tag{65}$$

$$B_{\theta} = B_r \frac{r}{U_{\rm sw}} \omega^* \sin \beta^* \sin \phi^* \tag{66}$$

$$B_{\phi} = B_{r} \frac{r}{U_{sw}} \left[ \omega^{*} \sin \beta^{*} \cos \theta \cos \phi^{*} + \sin \theta (\omega^{*} \cos \beta^{*} - \Omega_{\odot}) + \frac{d\omega^{*}}{d\theta} \sin \beta^{*} \sin \theta \cos \phi^{*} + \omega^{*} \frac{d\beta^{*}}{d\theta} \cos \beta^{*} \sin \theta \cos \phi^{*} \right].$$
(67)

Here

$$\phi^* = \phi - \Omega_{\odot} t + \frac{\Omega(r - r_0)}{U_{\rm sw}} + \phi_0.$$
(68)

 $B_0$  is again the radial component of the magnetic field at the reference radius  $r_0$ . The symbol  $\beta_{\rm F}$  is the angle (the Fisk angle) between the virtual magnetic axis (p-axis) and the rotation axis of the Sun, and  $\omega$ is the differential rotation rate of the Sun. Both the angle  $\beta_{\rm F}$  and  $\omega$  are generalized to the latitudinal dependent case by introducing the transition function  $F_{\rm t}(\theta)$  in the following way:

$$\beta^* = \beta_{\rm F} F_{\rm t}(\theta) \tag{69}$$

$$\omega^* = \omega F_{\rm t}(\theta). \tag{70}$$

The transition function is constructed as follows (Burger et al., 2008):

$$F_{\rm t} = \left| \tanh[\delta_{\rm pol}\theta] + \tanh[\delta_{\rm pol}(\theta - \pi)] - \tanh[\delta_{\rm eq}(\theta - \theta_{\rm b}')] \right|^2 \tag{71}$$

for the northern high-latitude region  $(0 \le \theta < \theta'_{\rm b});$

$$F_{\rm t} = 0 \tag{72}$$

for the equatorial or low-latitude region  $(\theta'_{\rm b} \leq \theta \leq \pi - \theta'_{\rm b})$ ; and

$$F_{\rm t} = \left| \tanh[\delta_{\rm pol}\theta] + \tanh[\delta_{\rm pol}(\theta - \pi)] - \tanh[\delta_{\rm eq}(\theta - \pi + \theta_{\rm b}')] \right|^2$$
(73)

for the southern high-latitude region.  $\theta'_{\rm b}$  is the equatorward-limit polar angle of the coronal hole (characterized by open field lines) and is between  $60^{\circ}$  and  $80^{\circ}$  from the solar rotation axis in Burger et al. (2008). The symbols  $\delta_{\rm pol}$  and  $\delta_{\rm eq}$  are the control parameters of the transition from the high-latitude magnetic fields (Fisk-type model) into the low-latitude fields (Parker-type model), e.g.,  $\delta_{\rm pol} = \delta_{\rm eq} = 5.0$  proposed by Burger et al. (2008). The magnetic field model in Eqs. (65)–(67) represent a natural extension of the Parker model in that the case  $F_t = 1$  reproduces the model proposed by Zurbuchen et al. (1997) and the case  $F_t = 0$  the Parker model. The associated polar and azimuthal components of the flow velocity are:

$$U_{\theta} = r_{0}\omega^{*} \sin\beta^{*} \sin\phi_{\Omega}$$

$$U_{\phi} = r_{0} \left( \omega^{*} \sin\beta^{*} \cos\theta \cos\phi_{\Omega} + \omega^{*} \cos\beta^{*} \sin\theta + \frac{d\omega}{d\theta} \sin\beta^{*} \sin\theta \cos\phi_{\Omega} + \omega^{*} \frac{d\beta^{*}}{d\theta} \sin\theta \cos\phi_{\Omega} + \omega^{*} \frac{d\beta^{*}}{d\theta} \sin\theta \cos\phi_{\Omega} \right).$$

$$(74)$$

The Fisk angle  $\beta_{\rm F}$  is related to the tile angle of the heliospheric current sheet  $\alpha_{\rm F}$  by Burger et al. (2008):

$$\cos\left(\alpha_{\rm F} + \beta_{\rm F}\right) = 1 - \left(1 - \cos\theta_{\rm mm}'\right) \frac{\sin^2 \alpha_{\rm F}}{\sin^2 \theta_{\rm mm}},\tag{76}$$

where  $\theta_{\rm mm}$  and  $\theta'_{\rm mm}$  are the equatorward (low-latitude) boundary of the polar coronal hole on the level of photosphere source surface in heliomagnetic coordinates, respectively. The boundary angles are expressed in heliographic coordinates as  $\theta_{\rm b} = \theta_{\rm mm} - \alpha_{\rm F}$  and  $\theta'_{\rm b} = \theta'_{\rm mm} - \alpha_{\rm F}$ , respectively.

The tilt angles  $\alpha_{\rm F}$  and  $\beta_{\rm F}$  and the boundary angles  $\theta_{\rm b}$  and  $\theta'_{\rm b}$  can be modeled in a time-dependent way when constructing the Fisk-Parker-hybrid model (Burger et al., 2008) as a solar cycle dependent one: The time dependence of the tilt angle  $\alpha_{\rm F}$  is modeled as

$$\alpha_{\rm F} = \alpha_{\rm min} + \left(\frac{\pi}{4} - \frac{\alpha_{\rm min}}{2}\right) \left[1 - \cos\left(\frac{\pi}{4}T[{\rm yr}]\right)\right] \tag{77}$$

for  $0 \leq T[yr] \leq 4yr$ , and

$$\alpha_{\rm F} = \alpha_{\rm min} + \left(\frac{\pi}{4} - \frac{\alpha_{\rm min}}{2}\right) \left[1 - \cos\left(\frac{\pi}{7}(T[{\rm yr}] - 11)\right)\right] \tag{78}$$

for  $4 < T \leq 11$ yr, where  $\alpha_{\min} = \pi/18$  is an offset tilt angle. Time T is measured in units of years after a solar minimum. The time dependence of the boundary angles is

$$\theta_{\rm b} = \frac{\theta_{\rm b(min)}}{2} \left[ 1 + \cos\left(\frac{\pi}{4}T[{\rm yr}]\right) \right]$$
(79)

$$\theta_{\rm b}' = \frac{\theta_{\rm b(min)}'}{2} \left[ 1 + \cos\left(\frac{\pi}{4}T[{\rm yr}]\right) \right]$$
(80)

for  $0 \leq T \leq 4$ yr, and

$$\theta_{\rm b} = \frac{\theta_{\rm b(min)}}{2} \left\{ 1 + \cos\left[\frac{\pi}{7}(T[{\rm yr}] - 11)\right] \right\}$$
(81)

$$\theta_{\rm b}' = \frac{\theta_{\rm b(min)}'}{2} \left\{ 1 + \cos\left[\frac{\pi}{7}(T[{\rm yr}] - 11)\right] \right\}$$
(82)

for  $4 < T \leq 11$  yr."

A more detailed explanation of the two-dimensional MHD model by Sakurai (1985) is included in section 3.1, subsection "two-dimensional treatment". (page 19–20)

"It is useful to introduce the poloidal-toroidal expression of the magnetic field in the two-dimensional MHD treatment:

$$\vec{B} = \nabla \times (a\vec{e}_{\phi}) + B_{\phi}\vec{e}_{\phi}, \tag{90}$$

where a denotes the magnetic stream function and  $\vec{e}_{\phi}$  is the unit vector in the azimuthal direction around the rotation axis. The poloidal fields  $B_{\rm p}$  (the first term in Eq. 90) are obtained by a family of curves under a = const. We introduce the barred radius which is the distance from the rotation axis,  $\bar{r} = r \sin \theta$ . The flow velocity is decomposed by referring to the local magnetic field as

$$\vec{U} = \frac{\alpha_{\rm m}(a)}{\rho} \vec{B} + \bar{r}^2 \Omega(a) \vec{e}_{\phi}, \qquad (91)$$

where the first term (denoted by  $U_{\rm p}$ ) is the flow velocity component parallel to the magnetic field in the frame rotating with the angular velocity  $\Omega$ , and the second term (denoted by  $U_{\phi}$ ) is perpendicular to the magnetic field. The toroidal component of magnetic field is determined by the angular momentum conservation,

$$\bar{r}\left(U_{\phi} - \frac{B_{\phi}}{\mu_0 a}\right) = l = \Omega \bar{r}_{\rm A}^2(a), \tag{92}$$

where l is the specific angular momentum and  $\bar{r}_A$  is the Alfvén radius at which the poloidal component of the flow velocity becomes equal to the Alfvén speed for the poloidal component of the magnetic field. Equation (92) is obtained from the (steady-state) MHD momentum equation and the flow velocity expression in Eq. (91). The magnetic stream function needs to be determined for the flow velocity and the poloidal component of the magnetic field. The magnetic stream function is numerically evaluated from the momentum equation (or force balance) perpendicular to the magnetic field by solving the following equation (Sakurai, 1985):

$$\nabla \cdot \left[ \left( \frac{\alpha_{\rm m}^2}{\rho} - \frac{1}{\mu_0} \right) \frac{\nabla a}{\bar{r}^2} \right] = \rho \left( E' - \frac{1}{\gamma_{\rm p} - 1} \frac{p}{\rho} \frac{K'}{K} + \bar{r}^2 \Omega \Omega' \right) + \frac{B_{\rm p}^2}{\rho} \alpha_{\rm m} \alpha_{\rm m}' + D \left[ \frac{D}{\mu_0} \Omega^2 \bar{r}^2 \alpha_{\rm m} \alpha_{\rm m}' - \alpha_{\rm m}^2 \Omega^2 (\bar{r}_{\rm A}^2)' - \alpha_{\rm m}^2 \Omega \Omega' (\bar{r}_{\rm A}^2 - \bar{r}_{\rm A}) \right],$$

where

$$D = \frac{\mu_0 \rho \left(\bar{r}_{\rm A}^2 - r^2\right)}{\bar{r}^2 (\mu_0 \rho \alpha_{\rm m}^2 - \rho)}$$
(94)

and the prime  $(\cdot)'$  denotes the differentiation with respect to the magnetic stream function, d/da. Equation (93) is the generalized Grad-Shafranov

equation for the two-dimensional centrifugally-driven wind. The density  $\rho$  follows the Bernoulli equation:

$$\frac{U_{\rm p}^2}{2} + \frac{1}{2}(U_{\phi} - \Omega\bar{r})^2 + \frac{\gamma_{\rm p}}{\gamma_{\rm p} - 1}\frac{p}{\rho} - \frac{GM}{r} - \frac{\Omega^2\bar{r}^2}{2} = E(a)$$
(95)

under the polytropic or adiabatic equation of state

$$p = K(a)\rho^{\gamma_{\rm P}}.\tag{96}$$

In the two-dimensional MHD treatment of the flow, the wind becomes collimated toward the rotation axis by the pinch of toroidal fields (Sakurai, 1985), causing a non-zero poleward (northward or southward) component of the magnetic field.

 A more detailed explanation about the effect of turbulent diffusion and a model construction for the turbulent diffusion are added to section 3.3, subsection "Turbulent diffusion". (page 20–22)

"Turbulence on smaller spatial scales serves as an energy sink to largescale mean fields, which leads to the notion of turbulent diffusion (meanfield electrodynamics). To see this more clearly, one may decompose the magnetic field into a large-scale mean field  $\vec{B}_0$  and a fluctuating field  $\delta \vec{B}$ (with the zero mean value); and the flow velocity likewise:

$$\vec{B} = \vec{B}_0 + \delta \vec{B} \tag{97}$$

$$\vec{U} = \vec{U}_0 + \delta \vec{U}. \tag{98}$$

The induction equation for the large-scale magnetic field has then the frozen-in term for the large-scale fields  $\vec{B}_0$  and  $\vec{U}_0$  and the electromotive force term  $\mathcal{E}_{em}$ :

$$\frac{\partial \vec{B}_0}{\partial t} = \nabla \times \left( \vec{U}_0 \times \vec{B}_0 \right) + \nabla \times \vec{\mathcal{E}}_{em}.$$
(99)

The electromotive force is an averaged electric field coming from the coupling of the fluctuating with the fluctuating magnetic field by the cross product:

$$\vec{\mathcal{E}}_{\rm em} = \left\langle \delta \vec{U} \times \delta \vec{B} \right\rangle. \tag{100}$$

A widely-used model in the mean-field electrodynamics is that the electromotive force depends on the large-scale quantities such as the largescale magnetic field, the curl of the large-scale magnetic field, and the curl of the large-scale flow velocity. By introducing the proper transport coefficients  $\alpha_t$ ,  $\beta_t$ , and  $\gamma_t$ , the electromotive force is modeled as

$$\vec{\mathcal{E}}_{\text{model}} = \alpha_{\text{t}} \vec{B}_0 - \beta_{\text{t}} \nabla \times \vec{B}_0 + \gamma_{\text{t}} \nabla \times \vec{U}_0.$$
(101)

After some algebra using Eqs. (99) and (100), one identifies that the term  $\beta_t \nabla \times \vec{B}_0$  becomes nothing other than the diffusion term for the

large-scale magnetic field (under the condition that the coefficient  $\beta_t$  is not negative):

\_

$$\frac{\partial \vec{B}_0}{\partial t} = \nabla \times \left( \vec{U}_0 \times \vec{B}_0 \right) + \nabla \times \left( \alpha_t \vec{B}_0 \right) + \beta_t \nabla^2 \vec{B}_0 + \nabla \times \left( \gamma_t \nabla \times \vec{U}_0 \right).$$
(102)

The terms with  $\alpha_t$  and  $\gamma_t$  in turn may amplify the large-scale magnetic field when the coefficients are in favor of field amplification (dynamo mechanism). The transport coefficients are theoretically estimated as follows:

$$\alpha_{\rm t} = C_{\alpha} \tau (-h_{\rm kin} + h_{\rm cur}) \tag{103}$$

$$\beta_{\rm t} = C_{\beta}\tau \left(e_{\rm kin} + e_{\rm mag}\right) \tag{104}$$

$$\gamma_{\rm t} = C_{\gamma} \tau h_{\rm crs}, \qquad (105)$$

where  $C_{\alpha}$ ,  $C_{\beta}$ , and  $C_{\gamma}$  are dimensionless scalar factors, and are estimated as (Yoshizawa, 1998),

$$C_{\alpha} \simeq 0.02 \tag{106}$$

$$C_{\beta} \simeq 0.05 \tag{107}$$

$$C_{\gamma} \simeq 0.04. \tag{108}$$

The symbol  $\tau$  denotes the turbulent correlation time length, and h and e represent the helicity and the energy quantities:  $h_{\rm kin}$  the kinetic helicity density,  $h_{\rm cur}$  the current helicity density,  $h_{\rm crs}$  the cross helicity density,  $e_{\rm kin}$  the turbulent kinetic energy density, and  $e_{\rm mag}$  the turbulent magnetic energy density. The helicity density quantities and the energy density quantities are defined for the fluctuating field,

$$h_{\rm kin} = \left\langle \delta \vec{U} \cdot \left( \nabla \times \delta \vec{U} \right) \right\rangle \tag{109}$$

$$h_{\rm cur} = \frac{1}{\mu_0 \rho_0} \left\langle \delta \vec{B} \cdot \left( \nabla \times \delta \vec{B} \right) \right\rangle \tag{110}$$

$$h_{\rm crs} = \frac{1}{\sqrt{\mu_0 \rho_0}} \left\langle \delta \vec{U} \cdot \delta \vec{B} \right\rangle \tag{111}$$

$$e_{\rm kin} = \frac{1}{2} \langle |\delta \vec{U}|^2 \rangle \tag{112}$$

$$e_{\text{mag}} = \frac{1}{2\mu_0\rho_0} \langle |\delta\vec{B}|^2 \rangle.$$
(113)

Note that different definitions are possible for the helicity and energy density quantities. In the definition above (Eqs. 109–113) the fluctuating magnetic field is converted into the velocity dimension such as  $\delta \vec{B}/\sqrt{\mu_0\rho_0}$  and the energy density is represented as that per unit mass. The correlation time length  $\tau$  can in the simplest case be modeled or represented by the eddy turnover time,

$$\tau_{\rm ed} = \frac{\ell}{\delta U} = \frac{e_{\rm kin} + e_{\rm mag}}{\varepsilon},\tag{114}$$

where  $\varepsilon$  is the dissipation rate which needs to be obtained by solving an equation in the similar fashion to the turbulence energy (Yokoi, 2008). The estimate of time scale can be extended by including the Alfvén time

effect into a synthesized time scale  $\tau_{\rm s}$  in the additive sense in the frequency domain as

$$\frac{1}{\tau_{\rm s}} = \frac{1}{\tau_{\rm ed}} + \chi \frac{1}{\tau_{\rm A}},\tag{115}$$

where  $\tau_{\rm A}$  denotes the Alfvén time

$$\tau_{\rm A} = \frac{\ell}{V_{\rm A}} = \frac{|e_{\rm kin} + e_{\rm mag}|^2}{\varepsilon V_{\rm A}^2} \tag{116}$$

with the length scale  $\ell$  and the Alfvén speed  $V_{\rm A}$ . The symbol  $\chi$  is the weight factor for the Alfvén time, and is estimated to be of the order  $10^2$  in the solar wind application (Yokoi, 2008). A more rigorous treatment is to solve two sets of equations, one for the large-scale mean fields and the other for the small-scale turbulent fields. This task can be achieved either analytically using the two-scale direct interaction approximation (Yokoi, 2006; Yokoi and Hamba, 2007; Yokoi et al., 2008) or numerically (Usmanov et al., 2012, 2014, 2016)."

Subsection "Pickup ions" is extended by showing model equations. (page 22–23)

"Pickup ions from interstellar neutral hydrogen atoms are one of the ingredients to the solar wind, and contribute to additional mass of the plasma, which results in deceleration of the solar wind expansion and in increase in the plasma temperature. Pickup ions originate in (1) charge exchange with the solar wind protons and (2) photoionization by the solar radiation. Steady-state MHD equations for the wind including pickup ions are introduced by Isenberg (1986) and Whang (1998), and are numerically implemented to simulation studies for a three-component fluid (thermal protons, electrons, pickup protons) by Usmanov and Goldstein (2006); Usmanov et al. (2014) and for a four-component fluid by adding interstellar hydrogen (Usmanov et al., 2016).

The continuity equation in the one-fluid sense (mixture of electrons, solar wind protons, and pickup ions of interstellar origin) has a contribution from the photoionization as a source term. and is written for the steady state as (Whang, 1998)

$$\nabla \cdot (\rho \vec{U}) = m_{\rm p} q_{\rm ph},\tag{117}$$

where  $\rho$  and  $\vec{U}$  denote the mass density and the flow velocity in the onefluid sense,  $m_{\rm p}$  the proton mass, and  $q_{\rm ph}$  the pickup ion production rate by the photoionization process,

$$q_{\rm ph} = \nu_0 \left(\frac{r_0^2}{r}\right) n_{\rm nt}.$$
 (118)

Here  $\nu_0 = 0.9 \times 10^{-7} \text{ s}^{-1}$  is the photoionization rate per hydrogen atom at the Earth orbit distance as reference  $r_0 = 1$  au, and  $n_{\text{nt}}$  is the number density of neutral hydrogen (of interstellar origin). The one-fluid momentum equation in the steady state is approximated into (by neglecting higher-order terms) (Whang, 1998)

$$\rho \vec{U} \cdot \vec{U} + \nabla P - \rho \nabla \left(\frac{GM_{\odot}}{r}\right) - \frac{1}{\mu_0} (\nabla \times \vec{B}) \times \vec{B} = -(q_{\rm ex} + q_{\rm ph}) m_{\rm p} \vec{U}.$$
(119)

Here  $q_{\rm ex}$  is the pickup ion production rate by the charge exchange process,

$$q_{\rm ex} = \sigma_{\rm ex} n_{\rm sw} n_{\rm nt} U, \tag{120}$$

where  $\sigma_{\text{ex}}$  is the cross section of charge exchange between a hydrogen atom and the solar wind protons,  $n_{\text{sw}}$  is the number density of solar wind protons."

 Section 3.3 (Stellar wind and interstellar space) is extended by referring to the models by Johnstone et al. (2015), Keppens and Goedlbloed (1999), Thirumalai and Heyl (2010), and Kriticka et al. (2016). (page 23)

"Various outflow models have been proposed for the stellar wind. For example, a wind model is constructed and numerically studied for the thermallly-driven hydrodynamic outflow from low-mass stars (Johnstone et al., 2015). A dead zone due to the magnetic dipole field effect can arise in the equatorial region (Keppens and Goeldbloed, 1999). A model is also constructed for the stellar winds around asymptotic giant branch (AGB) stars with dust grains by employing the MHD equation for the stellar wind plasma and the Euler equation for the dust grains under the gravity, the radiation pressure, and the drag force (Thirumalai and Heyl, 2010), showing the possibility of a stellar wind driven by dust grains. Mass-loss rate is observationally studied via stellar winds for subluminous stars (Krtička et al., 2016), in which the following flow velocity model is used for fitting with three parameters  $U_1$ ,  $U_2$ , and  $\gamma_{sw}$ :

$$U = \left[ U_1 \left( 1 - \frac{R_s}{r} \right) + U_2 \left( 1 - \frac{R_s}{r} \right)^2 \right] \left\{ 1 - \exp\left[ \gamma_{sw} \left( 1 - \frac{r}{R_s} \right)^2 \right] \right\},$$
(121)

where  $R_{\rm s}$  is the stellar radius."

A paragraph is added at the end of section 4 (Summary and conclusions) on page 25. We add only one paragraph in section 4 to keep the manuscript concise.

"It is also worth noting the limits of the models. First, the magnetic fields are highly structures in the solar corona and at the solar surface. At some distance sufficiently close to the Sun, the interplanetary magnetic field should smoothly be connected to the coronal magnetic field. Second, the outer heliosphere has the termination shock and the heliopause, which are not included in the models in this review. Third, the solar variability includes not only the 11-year sunspot number variation or the 22-year magnetic structure variation, but also modulations of the solar cycle on long time scales such as 100 or even 1000 years."

• I had minor comments on figures and captions but the other referee has already discussed them in more detail than I was planning.

**Reply**: We went through the manuscript text check again. All changes are marked in blue in the revised manuscript.

**Other changes**

- Analysis and extension of the Parker model by Summers (1978,1982) are cited in section 2.1.1. (page 5)
- All equations in separate lines have the equation numbers.
- Mathematical symbols have been re-assigned by using capital letters, small letters, caligraphic letters, asterisk, subscripts, to avoid confusion. Also, a circle is used instead of "degree" for the units of angles.
- The following reference items are added.
  - Alazraki and Couturier, Astron. Astrophys., 1971.
  - Belcher, Astrophys. J., 1971.
  - Isenberg, J. Geophys. Res., 1986.
  - Johnstone et al., Astron. Astrophys., 2015.
  - Keppens and Goedlbloed, Astron. Astrophys., 1999.
  - Krticka et al., Astron. Astrophys., 2016.
  - Lima et al., Astron. Astrophys., 2001.
  - Summers, J. Inst. Maths. Applics, 1978.
  - Summers, Astrophys. J., 1982.
  - Thirumalai and Heyl, Mon. Not. R. Astron. Soc., 2010.
  - Yoshizawa, Hydrodynamic and Magnetohydrodynamic Turbulent Flows: Modelling and Statistical Theory, 1998.

---

## Author Response (AR1)

**Reply to referee comments**

We thank the both referees for their time, the efforts for working on our manuscript, and for raising helpful comments. We revise the manuscript according to the suggestions by the referees, and realize that the manuscript quality has indeed been improved very much and the manuscript is more friendly to the potential readers, who may be students or experts in other research fields. Detailed response is shown in the following. Changes in the revised manuscript are marked in blue.
* * *
**Referee 1**

- *This draft reviews basic concepts of the large-scale heliospheric magnetic fields mainly from the kinematic framework. I have the following (mostly minor) comments:*

  *1. Fig.2 The blue, red, and gray lines should be defined in the caption.*

  **Reply**: Done. (page 8, fig. 2 caption)

  – Fig. 2 caption (page 8):
  "Streamlines in the Parker spiral model of interplanetary magnetic field around the Sun (a filled circle in yellow) in the heliospheric ecliptic plane up to 5 astronomical units (au) under different conditions of the solar wind speed. The orbit of the Earth is marked by a blue curve at a radius of 1 au, that of Mars by a red curve (1.5 au), and that of Jupiter by a green curve (5 au)."

- *2. The caption of Fig. 3 The authors should explain the line types and the numbers (24.47h, etc.: Rotational period) in the caption or in the panel.*

  **Reply**: Agreed. Done (page 9, fig. 3 caption)

  – Fig. 3 caption (page 9):
  "Heliocentric distance r in astronomical units (au) at which the spiral angle of the interplanetary magnetic field reaches 45° to the radial direction from the Sun ($B_r = B_\phi$ ). The curves are plotted as a function of the solar wind speed in units of km s$^{-1}$ for 3 different rotation rates, a period of 26.24 hours (upper curve), 25.38 hours (middle curve), and 24.47 hours (lower curve). A typical value of the solar wind speed is 430 km s$^{-1}$ (shown by a vertical thin line).

- *3. Fig. 4 and Section 2.1.4 In the section 2.1.4, it seems that the authors focuses on the situation, in which B_{\theta} = 0 (eq.20). But Figure 4 shows field lines with non-zero B_{\theta}. Probably, the formulation of eqs. (22) - (25) in the same section takes into account B_{\theta}. I think more explanations are necessary, which is friendly to readers.*

  **Reply**:

We agree that more explanation is necessary to clarity the polar component issue. The axial component of the spiral (or helical) field lines in Fig. 4 is due to the radial component, and does not represent the polar component. The vanishing polar component of the magnetic field holds in the Parker model; the polar component $B_\theta$ has the axial component like the radial component, but the polar component differs from the radial one in that the polar component is pointing toward the rotation axis (whereas the radial component is pointing away from the rotation axis). The radial direction and the polar direction are orthogonal to each other.

We add the following text and changed the figure 4 caption.

  – Main text, section 2.1.4 (page 10, line 187):

   "It is worth mentioning that the spiral magnetic field lines are constructed with the radial component from the Sun and the azimuthal component around the rotation axis, and do not contain the polar component (in the direction toward the rotation axis and perpendicular to the radial direction) as in Eqs. (28)–(30). The Parker spiral field lines have an axial component along the rotation axis but this is due to the radial component of the field line which has the axial component."

  – Figure 4 caption (page 11):

   "Note that the spiral magnetic field lines are constructed with the radial component from the Sun and the azimuthal component around the rotation axis, and do not contain the polar component (in the direction toward the rotation axis and perpendicular to the radial direction). The spiral field lines have an axial component along the rotation axis but this is due to the radial component of the spiral field line (in the sense of being away from the rotation axis)."

- *4. eq.(49) The scaling should be $r^{-1}$, instead of r. (I think this is simply a typo.)*

  **Reply**: Right! Thank you. (page 18, Eq. 86)

   – $B_\phi \propto r^{-1}$.

- *5. eq.(50) It is probably better to refer to old works (Alazraki & Couturier 1971; Belcher 1971), in addition to the recent works that are already cited in the present paper.*

  **Reply**: Agreed. Done. (page 18, line 370–371)

**Referee 2**

- *The article is a very nice tutorial overview of the subject. The grammar and spelling need to be reviewed, an example: page 3, line 19 "useful took" presumably should be "useful tool". I will leave this for the editorial staff and authors to go through this instead of providing an incomplete list.*

  **Reply**: Thank you for the positive evaluation and a careful check of the manuscript text.

  – "useful took" was corrected into "useful tool" (page 4, line 94).
  – We went through the spelling check and the sentence check to eliminate errors in English.

- *For the physics discussion, last section should really be expanded a little more to be a review rather than a tutorial. I would like to see a little more discussion of the two dimensional treatment, turbulent diffusion, as well as pickup ion effects on pages 16 and 17. A comprehensive review should include a basic discussion of the models. Authors already have a lot of the references in there. Including the model equations and a basic discussion of how they incorporate the higher order effects would make this review a good one stop overview read. I would like to see an expanded section 3.*

  **Reply**: Agreed. We added the following text and explanations.

  – A model including the latitude dependence (Lima et al., 2001) is added to section 2.2.1. (page 12, line 223–232).

  "A model of latitudinal dependence of the magnetic field is constructed by employing the method of separation of variable for an axi-symmetric magnetohydrodynamic outflow (Lima et al., 2001). The radial and the azimuthal components of the magnetic field are proposed as

  $$B_r = \frac{B_0}{r^2}\sqrt{1 + \mu \sin^{2\epsilon}\theta} \tag{41}$$

  $$B_\phi = \lambda B_0 \frac{\sin^\epsilon \theta}{r} \left( \frac{\frac{r^2}{R_s^2} - 1}{1 - M_A^2} \right), \tag{42}$$

  where $\epsilon$ is a free parameter, $\mu$ is the ratio of the flow kinetic energy (or energy density, strictly speaking) in the equatorial region to that in the polar region, and $\lambda$ is the ratio of azimuthal to radial velocity (and also magnetic field) at the base of the wind. $R_s$ is the radius of the star or the Sun. $M_A$ is the Alfvén Mach number of the flow. The polar component of the magnetic field is assumed to vanish due to the assumption of the axial symmetry around the rotation axis."
* * *
  – A model including the tilt angle and the solar cycle dependence (Burger et al., 2008) is added to section 2.2.4. (page 16, line 308 to page 18, line 344).

"A more refined magnetic field model is constructed by Burger et al. (2008), which offers an extension of the tilted heliospheric current sheet (with respect to the rotation axis) to the solar cycle dependence. The latitude-dependent magnetic field model is expressed as follows:

$$B_r = B_0 \left(\frac{r_0}{r}\right)^2 \tag{65}$$

$$B_\theta = B_r \frac{r}{U_{\text{sw}}} \omega^* \sin\beta^* \sin\phi^* \tag{66}$$

$$B_\phi = B_r \frac{r}{U_{\text{sw}}} \left[\omega^* \sin\beta^* \cos\theta \cos\phi^* + \right.$$
$$\sin\theta \left(\omega^* \cos\beta^* - \Omega_\odot\right) + \frac{\mathrm{d}\omega^*}{\mathrm{d}\theta} \sin\beta^* \sin\theta \cos\phi^* +$$
$$\left. \omega^* \frac{\mathrm{d}\beta^*}{\mathrm{d}\theta} \cos\beta^* \sin\theta \cos\phi^* \right]. \tag{67}$$

Here

$$\phi^* = \phi - \Omega_\odot t + \frac{\Omega(r - r_0)}{U_{\text{sw}}} + \phi_0. \tag{68}$$

$B_0$ is again the radial component of the magnetic field at the reference radius $r_0$. The symbol $\beta_{\text{F}}$ is the angle (the Fisk angle) between the virtual magnetic axis (p-axis) and the rotation axis of the Sun, and $\omega$ is the differential rotation rate of the Sun. Both the angle $\beta_{\text{F}}$ and $\omega$ are generalized to the latitudinal dependent case by introducing the transition function $F_{\text{t}}(\theta)$ in the following way:

$$\beta^* = \beta_{\text{F}} F_{\text{t}}(\theta) \tag{69}$$
$$\omega^* = \omega F_{\text{t}}(\theta). \tag{70}$$

The transition function is constructed as follows (Burger et al., 2008):

$$F_{\text{t}} = \left|\tanh[\delta_{\text{pol}}\theta] + \tanh[\delta_{\text{pol}}(\theta - \pi)] - \tanh[\delta_{\text{eq}}(\theta - \theta'_{\text{b}})]\right|^2 \tag{71}$$

for the northern high-latitude region ($0 \leq \theta < \theta'_{\text{b}}$);

$$F_{\text{t}} = 0 \tag{72}$$

for the equatorial or low-latitude region ($\theta'_{\text{b}} \leq \theta \leq \pi - \theta'_{\text{b}}$); and

$$F_{\text{t}} = \left|\tanh[\delta_{\text{pol}}\theta] + \tanh[\delta_{\text{pol}}(\theta - \pi)] - \tanh[\delta_{\text{eq}}(\theta - \pi + \theta'_{\text{b}})]\right|^2 \tag{73}$$

for the southern high-latitude region. $\theta'_{\text{b}}$ is the equatorward-limit polar angle of the coronal hole (characterized by open field lines) and is between $60°$ and $80°$ from the solar rotation axis in Burger et al. (2008). The symbols $\delta_{\text{pol}}$ and $\delta_{\text{eq}}$ are the control parameters of the transition from the high-latitude magnetic fields (Fisk-type model) into the low-latitude fields (Parker-type model), e.g., $\delta_{\text{pol}} = \delta_{\text{eq}} = 5.0$ proposed by Burger et al. (2008). The magnetic field model in Eqs. (65)–(67) represent a natural extension of the Parker model in that the case $F_{\text{t}} = 1$ reproduces the model proposed by Zurbuchen et al. (1997) and the case $F_{\text{t}} = 0$ the

Parker model. The associated polar and azimuthal components of the flow velocity are:

$$U_\theta = r_0 \omega^* \sin\beta^* \sin\phi_\Omega \qquad (74)$$

$$U_\phi = r_0 \bigg( \omega^* \sin\beta^* \cos\theta \cos\phi_\Omega + \omega^* \cos\beta^* \sin\theta +$$

$$\frac{\mathrm{d}\omega}{\mathrm{d}\theta} \sin\beta^* \sin\theta \cos\phi_\Omega +$$

$$\omega^* \frac{\mathrm{d}\beta^*}{\mathrm{d}\theta} \sin\theta \cos\phi_\Omega \bigg). \qquad (75)$$

The Fisk angle $\beta_\mathrm{F}$ is related to the tile angle of the heliospheric current sheet $\alpha_\mathrm{F}$ by Burger et al. (2008):

$$\cos\left(\alpha_\mathrm{F} + \beta_\mathrm{F}\right) = 1 - \left(1 - \cos\theta'_\mathrm{mm}\right) \frac{\sin^2 \alpha_\mathrm{F}}{\sin^2 \theta_\mathrm{mm}}, \qquad (76)$$

where $\theta_\mathrm{mm}$ and $\theta'_\mathrm{mm}$ are the equatorward (low-latitude) boundary of the polar coronal hole on the level of photosphere source surface in helio-magnetic coordinates, respectively. The boundary angles are expressed in heliographic coordinates as $\theta_\mathrm{b} = \theta_\mathrm{mm} - \alpha_\mathrm{F}$ and $\theta'_\mathrm{b} = \theta'_\mathrm{mm} - \alpha_\mathrm{F}$, respectively.

The tilt angles $\alpha_\mathrm{F}$ and $\beta_\mathrm{F}$ and the boundary angles $\theta_\mathrm{b}$ and $\theta'_\mathrm{b}$ can be modeled in a time-dependent way when constructing the Fisk-Parker-hybrid model (Burger et al., 2008) as a solar cycle dependent one: The time dependence of the tilt angle $\alpha_\mathrm{F}$ is modeled as

$$\alpha_\mathrm{F} = \alpha_\mathrm{min} + \left(\frac{\pi}{4} - \frac{\alpha_\mathrm{min}}{2}\right) \left[1 - \cos\left(\frac{\pi}{4}T[\mathrm{yr}]\right)\right] \qquad (77)$$

for $0 \le T[\mathrm{yr}] \le 4\mathrm{yr}$, and

$$\alpha_\mathrm{F} = \alpha_\mathrm{min} + \left(\frac{\pi}{4} - \frac{\alpha_\mathrm{min}}{2}\right) \left[1 - \cos\left(\frac{\pi}{7}(T[\mathrm{yr}] - 11)\right)\right] \qquad (78)$$

for $4 < T \le 11\mathrm{yr}$, where $\alpha_\mathrm{min} = \pi/18$ is an offset tilt angle. Time $T$ is measured in units of years after a solar minimum. The time dependence of the boundary angles is

$$\theta_\mathrm{b} = \frac{\theta_\mathrm{b(min)}}{2} \left[1 + \cos\left(\frac{\pi}{4}T[\mathrm{yr}]\right)\right] \qquad (79)$$

$$\theta'_\mathrm{b} = \frac{\theta'_\mathrm{b(min)}}{2} \left[1 + \cos\left(\frac{\pi}{4}T[\mathrm{yr}]\right)\right] \qquad (80)$$

for $0 \le T \le 4\mathrm{yr}$, and

$$\theta_\mathrm{b} = \frac{\theta_\mathrm{b(min)}}{2} \left\{1 + \cos\left[\frac{\pi}{7}(T[\mathrm{yr}] - 11)\right]\right\} \qquad (81)$$

$$\theta'_\mathrm{b} = \frac{\theta'_\mathrm{b(min)}}{2} \left\{1 + \cos\left[\frac{\pi}{7}(T[\mathrm{yr}] - 11)\right]\right\} \qquad (82)$$

for $4 < T \le 11\mathrm{yr}$."
* * *
– A more detailed explanation of the two-dimensional MHD model by Saku-
rai (1985) is included in section 3.1, subsection "two-dimensional treat-
ment". (page 19–20)

"It is useful to introduce the poloidal-toroidal expression of the magnetic
field in the two-dimensional MHD treatment:

$$\vec{B} = \nabla \times (a\vec{e}_\phi) + B_\phi \vec{e}_\phi, \tag{90}$$

where $a$ denotes the magnetic stream function and $\vec{e}_\phi$ is the unit vector
in the azimuthal direction around the rotation axis. The poloidal fields
$B_\mathrm{p}$ (the first term in Eq. 90) are obtained by a family of curves under
$a = const$. We introduce the barred radius which is the distance from the
rotation axis, $\bar{r} = r\sin\theta$. The flow velocity is decomposed by referring to
the local magnetic field as

$$\vec{U} = \frac{\alpha_\mathrm{m}(a)}{\rho}\vec{B} + \bar{r}^2\Omega(a)\vec{e}_\phi, \tag{91}$$

where the first term (denoted by $U_\mathrm{p}$) is the flow velocity component par-
allel to the magnetic field in the frame rotating with the angular velocity
$\Omega$, and the second term (denoted by $U_\phi$) is perpendicular to the mag-
netic field. The toroidal component of magnetic field is determined by
the angular momentum conservation,

$$\bar{r}\left(U_\phi - \frac{B_\phi}{\mu_0 a}\right) = l = \Omega\bar{r}_\mathrm{A}^2(a), \tag{92}$$

where $l$ is the specific angular momentum and $\bar{r}_\mathrm{A}$ is the Alfvén radius at
which the poloidal component of the flow velocity becomes equal to the
Alfvén speed for the poloidal component of the magnetic field. Equation
(92) is obtained from the (steady-state) MHD momentum equation and
the flow velocity expression in Eq. (91). The magnetic stream function
needs to be determined for the flow velocity and the poloidal component
of the magnetic field. The magnetic stream function is numerically eval-
uated from the momentum equation (or force balance) perpendicular to
the magnetic field by solving the following equation (Sakurai, 1985):

$$
\nabla \cdot \left[\left(\frac{\alpha_\mathrm{m}^2}{\rho} - \frac{1}{\mu_0}\right)\frac{\nabla a}{\bar{r}^2}\right] = \rho\left(E' - \frac{1}{\gamma_\mathrm{p} - 1}\frac{p}{\rho}\frac{K'}{K} + \bar{r}^2\Omega\Omega'\right) +
$$
$$
\frac{B_\mathrm{p}^2}{\rho}\alpha_\mathrm{m}\alpha_\mathrm{m}' +
$$
$$
D\left[\frac{D}{\mu_0}\Omega^2\bar{r}^2\alpha_\mathrm{m}\alpha_\mathrm{m}' - \alpha_\mathrm{m}^2\Omega^2(\bar{r}_\mathrm{A}^2)' - \alpha_\mathrm{m}^2\Omega\Omega'\left(\bar{r}_\mathrm{A}^2 - \bar{r}_\mathrm{A}^2\right)\right] \tag{93}
$$

where

$$D = \frac{\mu_0\rho\left(\bar{r}_\mathrm{A}^2 - r^2\right)}{\bar{r}^2(\mu_0\rho\alpha_\mathrm{m}^2 - \rho)} \tag{94}$$

and the prime $(\cdot)'$ denotes the differentiation with respect to the magnetic
stream function, $\mathrm{d}/\mathrm{d}a$. Equation (93) is the generalized Grad-Shafranov

equation for the two-dimensional centrifugally-driven wind. The density $\rho$ follows the Bernoulli equation:

$$\frac{U_{\mathrm{p}}^2}{2} + \frac{1}{2}(U_\phi - \Omega\bar{r})^2 + \frac{\gamma_{\mathrm{p}}}{\gamma_{\mathrm{p}}-1}\frac{p}{\rho} - \frac{GM_\cdot}{r} - \frac{\Omega^2\bar{r}^2}{2} = E(a) \qquad (95)$$

under the polytropic or adiabatic equation of state

$$p = K(a)\rho^{\gamma_{\mathrm{p}}}. \qquad (96)$$

In the two-dimensional MHD treatment of the flow, the wind becomes collimated toward the rotation axis by the pinch of toroidal fields (Sakurai, 1985), causing a non-zero poleward (northward or southward) component of the magnetic field.
* * *
– A more detailed explanation about the effect of turbulent diffusion and a model construction for the turbulent diffusion are added to section 3.3, subsection "Turbulent diffusion". (page 20–22)

"Turbulence on smaller spatial scales serves as an energy sink to large-scale mean fields, which leads to the notion of turbulent diffusion (mean-field electrodynamics). To see this more clearly, one may decompose the magnetic field into a large-scale mean field $\vec{B}_0$ and a fluctuating field $\delta\vec{B}$ (with the zero mean value); and the flow velocity likewise:

$$\begin{aligned} \vec{B} &= \vec{B}_0 + \delta\vec{B} & (97) \\ \vec{U} &= \vec{U}_0 + \delta\vec{U}. & (98) \end{aligned}$$

The induction equation for the large-scale magnetic field has then the frozen-in term for the large-scale fields $\vec{B}_0$ and $\vec{U}_0$ and the electromotive force term $\mathcal{E}_{\mathrm{em}}$:

$$\frac{\partial\vec{B}_0}{\partial t} = \nabla \times \left(\vec{U}_0 \times \vec{B}_0\right) + \nabla \times \vec{\mathcal{E}}_{\mathrm{em}}. \qquad (99)$$

The electromotive force is an averaged electric field coming from the coupling of the fluctuating with the fluctuating magnetic field by the cross product:

$$\vec{\mathcal{E}}_{\mathrm{em}} = \left\langle \delta\vec{U} \times \delta\vec{B} \right\rangle. \qquad (100)$$

A widely-used model in the mean-field electrodynamics is that the electromotive force depends on the large-scale quantities such as the large-scale magnetic field, the curl of the large-scale magnetic field, and the curl of the large-scale flow velocity. By introducing the proper transport coefficients $\alpha_{\mathrm{t}}$, $\beta_{\mathrm{t}}$, and $\gamma_{\mathrm{t}}$, the electromotive force is modeled as

$$\vec{\mathcal{E}}_{\mathrm{model}} = \alpha_{\mathrm{t}}\vec{B}_0 - \beta_{\mathrm{t}}\nabla \times \vec{B}_0 + \gamma_{\mathrm{t}}\nabla \times \vec{U}_0. \qquad (101)$$

After some algebra using Eqs. (99) and (100), one identifies that the term $\beta_{\mathrm{t}}\nabla \times \vec{B}_0$ becomes nothing other than the diffusion term for the

large-scale magnetic field (under the condition that the coefficient $\beta_\mathrm{t}$ is not negative):

$$\frac{\partial \vec{B}_0}{\partial t} = \nabla \times \left(\vec{U}_0 \times \vec{B}_0\right) + \nabla \times \left(\alpha_\mathrm{t} \vec{B}_0\right) + \beta_\mathrm{t} \nabla^2 \vec{B}_0 + \nabla \times \left(\gamma_\mathrm{t} \nabla \times \vec{U}_0\right). \quad (102)$$

The terms with $\alpha_\mathrm{t}$ and $\gamma_\mathrm{t}$ in turn may amplify the large-scale magnetic field when the coefficients are in favor of field amplification (dynamo mechanism). The transport coefficients are theoretically estimated as follows:

$$\begin{align}
\alpha_\mathrm{t} &= C_\alpha \tau(-h_\mathrm{kin} + h_\mathrm{cur}) \tag{103}\\
\beta_\mathrm{t} &= C_\beta \tau \left(e_\mathrm{kin} + e_\mathrm{mag}\right) \tag{104}\\
\gamma_\mathrm{t} &= C_\gamma \tau h_\mathrm{crs}, \tag{105}
\end{align}$$

where $C_\alpha$, $C_\beta$, and $C_\gamma$ are dimensionless scalar factors, and are estimated as (Yoshizawa, 1998),

$$\begin{align}
C_\alpha &\simeq 0.02 \tag{106}\\
C_\beta &\simeq 0.05 \tag{107}\\
C_\gamma &\simeq 0.04. \tag{108}
\end{align}$$

The symbol $\tau$ denotes the turbulent correlation time length, and $h$ and $e$ represent the helicity and the energy quantities: $h_\mathrm{kin}$ the kinetic helicity density, $h_\mathrm{cur}$ the current helicity density, $h_\mathrm{crs}$ the cross helicity density, $e_\mathrm{kin}$ the turbulent kinetic energy density, and $e_\mathrm{mag}$ the turbulent magnetic energy density. The helicity density quantities and the energy density quantities are defined for the fluctuating field,

$$\begin{align}
h_\mathrm{kin} &= \left\langle \delta \vec{U} \cdot \left(\nabla \times \delta \vec{U}\right)\right\rangle \tag{109}\\
h_\mathrm{cur} &= \frac{1}{\mu_0 \rho_0} \left\langle \delta \vec{B} \cdot \left(\nabla \times \delta \vec{B}\right)\right\rangle \tag{110}\\
h_\mathrm{crs} &= \frac{1}{\sqrt{\mu_0 \rho_0}} \left\langle \delta \vec{U} \cdot \delta \vec{B}\right\rangle \tag{111}\\
e_\mathrm{kin} &= \frac{1}{2}\langle|\delta \vec{U}|^2\rangle \tag{112}\\
e_\mathrm{mag} &= \frac{1}{2\mu_0 \rho_0}\langle|\delta \vec{B}|^2\rangle. \tag{113}
\end{align}$$

Note that different definitions are possible for the helicity and energy density quantities. In the definition above (Eqs. 109–113) the fluctuating magnetic field is converted into the velocity dimension such as $\delta \vec{B}/\sqrt{\mu_0 \rho_0}$ and the energy density is represented as that per unit mass. The correlation time length $\tau$ can in the simplest case be modeled or represented by the eddy turnover time,

$$\tau_\mathrm{ed} = \frac{\ell}{\delta U} = \frac{e_\mathrm{kin} + e_\mathrm{mag}}{\varepsilon}, \quad (114)$$

where $\varepsilon$ is the dissipation rate which needs to be obtained by solving an equation in the similar fashion to the turbulence energy (Yokoi, 2008). The estimate of time scale can be extended by including the Alfvén time

effect into a synthesized time scale $\tau_{\mathrm{s}}$ in the additive sense in the frequency domain as

$$\frac{1}{\tau_{\mathrm{s}}} = \frac{1}{\tau_{\mathrm{ed}}} + \chi \frac{1}{\tau_{\mathrm{A}}}, \tag{115}$$

where $\tau_{\mathrm{A}}$ denotes the Alfvén time

$$\tau_{\mathrm{A}} = \frac{\ell}{V_{\mathrm{A}}} = \frac{|e_{\mathrm{kin}} + e_{\mathrm{mag}}|^2}{\varepsilon V_{\mathrm{A}}^2} \tag{116}$$

with the length scale $\ell$ and the Alfvén speed $V_{\mathrm{A}}$. The symbol $\chi$ is the weight factor for the Alfvén time, and is estimated to be of the order $10^2$ in the solar wind application (Yokoi, 2008). A more rigorous treatment is to solve two sets of equations, one for the large-scale mean fields and the other for the small-scale turbulent fields. This task can be achieved either analytically using the two-scale direct interaction approximation (Yokoi, 2006; Yokoi and Hamba, 2007; Yokoi et al., 2008) or numerically (Usmanov et al., 2012, 2014, 2016)."
* * *
– Subsection "Pickup ions" is extended by showing model equations. (page 22–23)

"Pickup ions from interstellar neutral hydrogen atoms are one of the ingredients to the solar wind, and contribute to additional mass of the plasma, which results in deceleration of the solar wind expansion and in increase in the plasma temperature. Pickup ions originate in (1) charge exchange with the solar wind protons and (2) photoionization by the solar radiation. Steady-state MHD equations for the wind including pickup ions are introduced by Isenberg (1986) and Whang (1998), and are numerically implemented to simulation studies for a three-component fluid (thermal protons, electrons, pickup protons) by Usmanov and Goldstein (2006); Usmanov et al. (2014) and for a four-component fluid by adding interstellar hydrogen (Usmanov et al., 2016).

The continuity equation in the one-fluid sense (mixture of electrons, solar wind protons, and pickup ions of interstellar origin) has a contribution from the photoionization as a source term. and is written for the steady state as (Whang, 1998)

$$\nabla \cdot (\rho \vec{U}) = m_{\mathrm{p}} q_{\mathrm{ph}}, \tag{117}$$

where $\rho$ and $\vec{U}$ denote the mass density and the flow velocity in the one-fluid sense, $m_{\mathrm{p}}$ the proton mass, and $q_{\mathrm{ph}}$ the pickup ion production rate by the photoionization process,

$$q_{\mathrm{ph}} = \nu_0 \left( \frac{r_0^2}{r} \right) n_{\mathrm{nt}}. \tag{118}$$

Here $\nu_0 = 0.9 \times 10^{-7} \ \mathrm{s}^{-1}$ is the photoionization rate per hydrogen atom at the Earth orbit distance as reference $r_0 = 1 \, \mathrm{au}$, and $n_{\mathrm{nt}}$ is the number density of neutral hydrogen (of interstellar origin). The one-fluid momentum equation in the steady state is approximated into (by neglecting

higher-order terms) (Whang, 1998)

$$\rho \vec{U} \cdot \vec{U} + \nabla P - \rho \nabla \left( \frac{GM_\odot}{r} \right) - \frac{1}{\mu_0} (\nabla \times \vec{B}) \times \vec{B} = -(q_{\mathrm{ex}} + q_{\mathrm{ph}}) m_{\mathrm{p}} \vec{U}. \quad (119)$$

Here $q_{\mathrm{ex}}$ is the pickup ion production rate by the charge exchange process,

$$q_{\mathrm{ex}} = \sigma_{\mathrm{ex}} n_{\mathrm{sw}} n_{\mathrm{nt}} U, \quad (120)$$

where $\sigma_{\mathrm{ex}}$ is the cross section of charge exchange between a hydrogen atom and the solar wind protons, $n_{\mathrm{sw}}$ is the number density of solar wind protons."
* * *
– Section 3.3 (Stellar wind and interstellar space) is extended by referring to the models by Johnstone et al. (2015), Keppens and Goedlbloed (1999), Thirumalai and Heyl (2010), and Kriticka et al. (2016). (page 23)

"Various outflow models have been proposed for the stellar wind. For example, a wind model is constructed and numerically studied for the thermallly-driven hydrodynamic outflow from low-mass stars (Johnstone et al., 2015). A dead zone due to the magnetic dipole field effect can arise in the equatorial region (Keppens and Goeldbloed, 1999). A model is also constructed for the stellar winds around asymptotic giant branch (AGB) stars with dust grains by employing the MHD equation for the stellar wind plasma and the Euler equation for the dust grains under the gravity, the radiation pressure, and the drag force (Thirumalai and Heyl, 2010), showing the possibility of a stellar wind driven by dust grains. Mass-loss rate is observationally studied via stellar winds for subluminous stars (Krtička et al., 2016), in which the following flow velocity model is used for fitting with three parameters $U_1$, $U_2$, and $\gamma_{\mathrm{sw}}$:

$$U = \left[ U_1 \left( 1 - \frac{R_{\mathrm{s}}}{r} \right) + U_2 \left( 1 - \frac{R_{\mathrm{s}}}{r} \right)^2 \right] \left\{ 1 - \exp \left[ \gamma_{\mathrm{sw}} \left( 1 - \frac{r}{R_{\mathrm{s}}} \right)^2 \right] \right\},$$
$$(121)$$

where $R_{\mathrm{s}}$ is the stellar radius."
* * *
– A paragraph is added at the end of section 4 (Summary and conclusions) on page 25. We add only one paragraph in section 4 to keep the manuscript concise.

"It is also worth noting the limits of the models. First, the magnetic fields are highly structures in the solar corona and at the solar surface. At some distance sufficiently close to the Sun, the interplanetary magnetic field should smoothly be connected to the coronal magnetic field. Second, the outer heliosphere has the termination shock and the heliopause, which are not included in the models in this review. Third, the solar variability includes not only the 11-year sunspot number variation or the 22-year magnetic structure variation, but also modulations of the solar cycle on long time scales such as 100 or even 1000 years."

- *I had minor comments on figures and captions but the other referee has already discussed them in more detail than I was planning.*

  **Reply**: We went through the manuscript text check again. All changes are marked in blue in the revised manuscript.
* * *
**Other changes**

- Analysis and extension of the Parker model by Summers (1978,1982) are cited in section 2.1.1. (page 5, line 133–136)

- All equations in separate lines have the equation numbers.

- Mathematical symbols have been re-assigned by using capital letters, small letters, caligraphic letters, asterisk, subscripts, to avoid confusion. Also, a circle is used instead of "degree" for the units of angles.

- The following reference items are added.

  - Alazraki and Couturier, Astron. Astrophys., 1971.
  - Belcher, Astrophys. J., 1971.
  - Isenberg, J. Geophys. Res., 1986.
  - Johnstone et al., Astron. Astrophys., 2015.
  - Keppens and Goedlbloed, Astron. Astrophys., 1999.
  - Krticka et al., Astron. Astrophys., 2016.
  - Lima et al., Astron. Astrophys., 2001.
  - Summers, J. Inst. Maths. Applics, 1978.
  - Summers, Astrophys. J., 1982.
  - Thirumalai and Heyl, Mon. Not. R. Astron. Soc., 2010.
  - Yoshizawa, Hydrodynamic and Magnetohydrodynamic Turbulent Flows: Modelling and Statistical Theory, 1998.

Manuscript prepared for J. Name
with version 4.2 of the LaTeX class copernicus.cls.
Date: 3 April 2019

[revised manuscript text omitted]

We rewrite Eqs. (13)–(15) into a simpler form as

$$B_r = B_0 \left(\frac{r_0}{r}\right)^2 \tag{17}$$

$$B_\phi = -r_0 B_0 \left(\frac{r_0}{r}\right) \frac{\Omega_\odot \cos\vartheta}{U_{\rm sw}}, \tag{18}$$

where, again, $B_0 = B(r_0, \theta, \phi_0)$ is the reference radial component of the magnetic field. (Meyer-Vernet, 2012).

We note that in Eqs. (17)–(18) the latitude $\vartheta$ (measured from the equator) is related to the polar angle $\theta$ (measured from the rotation axis) by $\theta = \pi - \vartheta$. By identifying or defining the radial and tangential components as $B_{\rm R} = B_r$ and $B_{\rm T} = B_\phi$, respectively, it is straightforward to transform the Parker spiral field into the RTN system as

$$B_R = B_0 \left(\frac{r_0}{r}\right)^2 \tag{19}$$

$$B_{\rm T} = -r_0 B_0 \left(\frac{r_0}{r}\right) \frac{\Omega_\odot \sin\theta}{U_{\rm sw}}. \tag{20}$$

Note that the normal component vanishes, $B_{\rm N} = 0$, because the Parker model does not include the polar component like the dipolar field of the Sun.

**2.1.3 Spiral angle**

The distance to the surface on which an azimuthal angle of $45°$ is realized (or $B_\theta \simeq B_r$) is approximately located at

$$r \simeq \frac{U_{\rm sw}}{\Omega_\odot} \sin\theta. \tag{21}$$

[Figure]

**Fig. 2.** Streamlines in the Parker spiral model of interplanetary magnetic field around the Sun (a filled circle in yellow) in the heliospheric ecliptic plane up to 5 astronomical units (au) under different conditions of the solar wind speed. The orbit of the Earth is marked by a blue curve at a radius of 1 au, that of Mars by a red curve (1.5 au), and that of Jupiter by a green curve (5 au).

Using the rotation period of the Sun 25.38 days (equivalent to an angular velocity of $\omega = 2.865 \times 10^{-6} \mathrm{rad\ s}^{-1}$) and the flow speed $U_{\mathrm{sw}} \simeq 430 \mathrm{\ km\ s}^{-1}$, the transition from the radially-dominant to the azimuthally-dominant magnetic field indeed happens around $r = 1$ au. The transition distance
170   is displayed as a function of the flow speed in Fig. 3 for three different solar rotation periods, 24.47 days, 25.38 days, and 26.24 days.

Alternatively, the Parker spiral model can be formulated in terms of the spiral angle $\psi$:

$$\tan\psi = \frac{\Omega_\odot(r - R_\odot)\sin\theta}{U_{\mathrm{sw}}},\qquad(22)$$

In this setting, the magnetic field $\boldsymbol{B}$ is, by using the unit vectors in the radial direction $\boldsymbol{e}_{\mathrm{r}}$ and in the azimuthal direction $\boldsymbol{e}_\phi$, given as

$$\boldsymbol{B} = B_0 \left(\frac{r_0}{r}\right)^2 (\boldsymbol{e}_{\mathrm{r}} - \tan\psi\, \boldsymbol{e}_\phi).\qquad(23)$$

In this formulation the magnitude of the magnetic field is estimated as

$$B = B_0 \left(\frac{r_0}{r}\right)^2 \sqrt{1 + \tan^2\psi}\qquad(24)$$

[Figure]

**Fig. 3.** Heliocentric distance $r$ in astronomical units (au) at which the spiral angle of the interplanetary magnetic field reaches $45°$ to the radial direction from the Sun ($B_r = B_\phi$). The curves are plotted as a function of the solar wind speed in units of km s$^{-1}$ for 3 different rotation rates, a period of 26.24 hours (upper curve), 25.38 hours (middle curve), and 24.47 hours (lower curve). A typical value of the solar wind speed is 430 km s$^{-1}$ (shown by a vertical thin line).

**2.1.4 Vector potential**

The magnetic vector potential $\boldsymbol{A}$ for the Parker spiral magnetic field under the Coulomb gauge $\nabla \cdot \boldsymbol{A} = 0$ can analytically be evaluated (Bieber et al., 1987). The vector potential in the following form,

$$A_r = \frac{2a\Omega_\odot}{3U_{\text{sw}}}\left(1 - \frac{3x}{2} - x\ln(1+x)\right) \tag{25}$$

$$A_\theta = \frac{2a\Omega_\odot}{3U_{\text{sw}}}\sin\theta\left(\frac{x}{1+x} + \ln(1+x)\right) \tag{26}$$

$$A_\phi = \frac{a}{r\sin\theta}(1-x), \tag{27}$$

where $x = |\cos\theta|$. Equations (25)–(27) correspond to the IMF in the following expression:

$$B_r = \frac{a}{r^2}\frac{\cos\theta}{|\cos\theta|} \tag{28}$$

$$B_\theta = 0 \tag{29}$$

$$B_\phi = -\frac{a\Omega_\odot}{U_{\text{sw}}}\frac{\sin\theta\cos\theta}{|\cos\theta|}. \tag{30}$$

Here $a$ is a free parameter proportional to the magnitude of the magnetic field in units of nT au$^2$ (for example, $a = 3.54$ nT au$^2$ produces a magnetic field of 5 nT at 1 au). The polar component of the

185 vector potential can be multiplied by a scalar function $f(\theta)$ to improve the accuracy of the model as $A_\theta \to f(\theta)A_\theta$.

Another formulation of the vector potential (again, under the Coulomb gauge) is to introduce a scalar potential as

$$\Phi_C = -\frac{2a\Omega_\odot r}{3u}\left(1 - \frac{3x}{2} - x\ln(1+x)\right), \tag{31}$$

which yields the following vector potential (Webb et al., 2010),

$$\boldsymbol{A} = a\left(\frac{1-|\cos\theta|}{r\sin\theta}\boldsymbol{e}_\phi - \frac{f(\theta)\Omega_\odot\sin\theta}{U_{\text{sw}}}\boldsymbol{e}_\theta\right). \tag{32}$$

Of course, in the both cases, Eqs. (25)–(27) and (32), the magnetic field is obtained by the definition of the vector potential as $\boldsymbol{B} = \nabla \times \boldsymbol{A}$. The electrostatic potential for the convective electric field $\boldsymbol{E} = -\boldsymbol{U} \times \boldsymbol{B} = -\nabla\Phi$ is

$$\Phi = -a\Omega_\odot\cos\theta. \tag{33}$$

The magnetic field lines for the Parker spiral model are shown in Fig. 4. Black lines have been calculated by the intersection of the two surfaces of constant Euler potentials $\alpha_E$, $\beta_E$ (Webb et al., 2010):

$$\alpha_E = -a|\cos\theta|, \quad \beta_E = \phi + \frac{\Omega_\odot r}{U_{\text{sw}}} - \Omega_\odot t. \tag{34}$$

It is worth mentioning that the spiral magnetic field lines are constructed with the radial component from the Sun and the azimuthal component around the rotation axis, and do not contain the polar component (in the direction toward the rotation axis and perpendicular to the radial direction) as
190 in Eqs. (28)–(30). The Parker spiral field lines have an axial component along the rotation axis but this is due to the radial component of the field line which has the axial component. For the sake of convenience one may set a value of unity to the variables $a$, $t$, $\Omega_\odot$, and $U_{\text{sw}}$ to provide the topology of the problem: $\alpha_E$ defines a cone (in green) that intersects a shell (in red) defined by $\beta_E$. Intersection lines define the magnetic field lines of the Parker model.

195 ## 2.2 Generalization of the Parker model

The Parker spiral model well approximates the mean, and large scale structure of the interplanetary magnetic field of our solar system. However, it fails to describe the three-dimensional geometry and evolution in time on various scales.

**2.2.1 Latitudinal dependence**

The Parker model does not recognize the sign reversal of the dipolar magnetic field over the north and the south hemispheres, the divergence-free nature of the magnetic field is not well represented. The hemispheric sign reversal can be incorporated into the Parker model as follows (Webb et al., 2010):

$$\boldsymbol{B} = \frac{af(\theta)}{r^2}\left(\boldsymbol{e}_r - \frac{\Omega_\odot r\sin\theta}{U_r}\boldsymbol{e}_\phi\right). \tag{35}$$

[Figure]

**Fig. 4.** Magnetic field lines (black curves) in the Parker spiral model for different latitude angles $\theta$ from the rotation axis. Curves are defined as the intersection of the surfaces of the Euler potentials, $\alpha_\mathrm{E} = \mathrm{const.}$ and $\beta_\mathrm{E} = \mathrm{const.}$, as presented by Webb et al. (2010). Note that the spiral magnetic field lines are constructed with the radial component from the Sun and the azimuthal component around the rotation axis, and do not contain the polar component (in the direction toward the rotation axis and perpendicular to the radial direction). The spiral field lines have an axial component along the rotation axis but this is due to the radial component of the spiral field line (in the sense of being away from the rotation axis).

Here, the constant $a$ and function $f = f(\theta)$ are given by:

$$a = \sigma_\mathrm{p} B_0 r_0^2 \tag{36}$$

$$f(\theta) = 1 - 2H(\theta - \pi/2) = \frac{\cos\theta}{|\cos\theta|}, \tag{37}$$

where $\sigma_\mathrm{p} = \pm 1$ defines the polarity of the magnetic field in the northern hemisphere of the sun, and $f(\theta)$ is the Heaviside step function with the property $f(\theta) = +1$ for $0 < \theta < \pi/2$ and $f(\theta) = -1$ for $\theta > \pi/2$.

A more elaborated analytic model is proposed along with the Ulysses measurements over the solar polar regions (Zurbuchen et al., 1997; Forsyth et al., 2002). The three-dimensional model

allows non-zero field in the polar component, and is expressed as

$$B_r = B_0 \left(\frac{r_0}{r}\right)^2 \tag{38}$$

$$B_\theta = \frac{B_0 r_0^2}{U_{\text{sw}} r} \, \omega \sin\beta_{\text{F}} \sin\left(\phi + \frac{r\Omega_\odot}{U_{\text{sw}}} - \phi_0\right) \tag{39}$$

$$B_\phi = -\frac{B_0 r_0^2}{U_{\text{sw}} r} \left[ \Omega_\odot \sin\theta - \omega \left( \cos\beta_{\text{F}} \sin\theta + \right. \right.$$

$$\left. \left. \sin\beta_{\text{F}} \cos\theta \cos\left(\phi + \frac{r\Omega_\odot}{U_{\text{sw}}} - \phi_0\right)\right)\right], \tag{40}$$

where $B_0$ is the radial component of magnetic field at the source surface located at heliospheric distance $r = r_0$, $\omega$ the differential rotation rate of the magnetic field line at foot points, $\beta_{\text{F}}$ (the Fisk angle) the polar angle at which a field line originating in the rotational pole crosses the source surface and is related to the angle between the solar magnetic dipole axis and the rotation axis, $\phi_0$ the heliographic longitude of the plane defined by the rotation and magnetic axes. The source magnetic field is defined at $r = r_0$. The angle $\phi = \phi_0$ occurs in the plane defined by the rotation axis and the magnetic axis of the Sun. Angle $\beta_{\text{F}}$ is the polar angle where the field line $p$ crosses the source surface (from the heliographic pole). The angle $\beta_{\text{F}}$ can be calculated in the model by Fisk (1996) for a given orientation $\alpha_{\text{F}}$ of the magnetic axis $M$ and a given non-radial expansion. For the configuration discussed by Fisk (1996), the value of $\beta_{\text{F}}$ is about $30°$.

A model of latitudinal dependence of the magnetic field is constructed by employing the method of separation of variable for an axi-symmetric magnetohydrodynamic outflow (Lima et al., 2001). The radial and the azimuthal components of the magnetic field are proposed as

$$B_r = \frac{B_0}{r^2} \sqrt{1 + \mu \sin^{2\epsilon}\theta} \tag{41}$$

$$B_\phi = \lambda B_0 \frac{\sin^\epsilon\theta}{r} \left(\frac{\frac{r^2}{R_s^2} - 1}{1 - M_{\text{A}}^2}\right), \tag{42}$$

where $\epsilon$ is a free parameter, $\mu$ is the ratio of the flow kinetic energy (or energy density, strictly speaking) in the equatorial region to that in the polar region, and $\lambda$ is the ratio of azimuthal to radial velocity (and also magnetic field) at the base of the wind. $R_s$ is the radius of the star or the Sun. $M_{\text{A}}$ is the Alfvén Mach number of the flow. The polar component of the magnetic field is assumed to vanish due to the assumption of the axial symmetry around the rotation axis.

**2.2.2 Poleward component**

The IMF can have a non-zero polar (or latitudinal) component, e.g., from the solar dipolar field. Generalization of the Parker model to the non-zero polar component case ($B_\theta \neq 0$) is based on the analysis by Forsyth et al. (1996). Let $\phi_B$ be the azimuthal angle that the projection of the IMF vector onto the R–T plane makes with the R axis in the right-handed sense, and $\delta_B$ be the meridional angle

of the IMF to the the R–T plane. These angles are defined in terms of the magnetic field components (Forsyth et al., 1996):

240 $\tan \phi_B = B_T/B_R$

$$\sin \delta_B = B_N/B, \tag{43}$$

where $B = \sqrt{B_R^2 + B_T^2 + B_N^2}$.

The azimuthal angle of the spiral field $\phi_P$ that the tangent to the ideal Parker spiral magnetic field makes with the radially outward direction at a position in interplanetary space specified by radial position $r$ and heliographic latitude $\delta$ is then given by :

$$\tan \phi_P = \frac{U_\phi - \Omega r \cos \delta}{U_r}. \tag{44}$$

On the assumption that $U_\phi$ is small, $\phi_P$ turns out to be negative. A magnetic field with a direction in agreement with the Parker spiral model will have either $\phi_B = \phi_P$ in a region of outward polarity or $\phi_B = 180° + \phi_P$ in a region of inward polarity field. In both regions the Parker model predicts that an ideal magnetic field has a meridional angle $\delta_B = 0°$ with respect to the R–T plane. Therefore, up to the second order in $B_N$ the sine of the meridional angle $\delta_B$ according to the second equation in Eq. (43) is given by

$$\sin \delta_B \simeq \frac{B_N}{\sqrt{B_R^2 + B_T^2}}. \tag{45}$$

If we combine the first of Eq. (43) together with Eq. (45) and solve for $B_T$ and $B_N$ we find up to $\mathcal{O}(B_N^3)$:

245 $$B_T = -B_0 \left(\frac{r_0}{r}\right)^2 \frac{(U_\phi - r\Omega_\odot \cos \delta)}{U_r} \tag{46}$$

$$B_N = B_0 \left(\frac{r_0}{r}\right)^2 \sqrt{1 + \frac{(U_\phi - r\Omega_\odot \cos \delta)^2}{U_r^2}} \sin \delta_B, \tag{47}$$

where we substituted $B_R$ by $B_r$ in Eqs. (17)–(18),

$$B_R = B_0 \left(\frac{r_0}{r}\right)^2. \tag{48}$$

Equations (46)–(48) provide a type of the Parker spiral magnetic field with the generalization to a non-zero normal component $B_N \neq 0$ parameterized by $\delta$ and $\delta_B$. For $\delta_B = 0°$ and ignoring the azimuthal component of the solar wind $U_\phi$, the model reproduces the Parker model, i.e., Eqs. (17)–

250 (18):

$$B_R = B_0 \left(\frac{r_0}{r}\right)^2 \tag{49}$$

$$B_T = -\left(\frac{r_0}{r}\right) r_0 B_0 \Omega_\odot \cos \delta \tag{50}$$

$$B_N = 0. \tag{51}$$

Another way of generalization is to use the power-law dependence using the power-law index $\kappa$ as a free parameter (Lhotka et al., 2016),

$$B_{\text{R}} = B_{\text{R0}} \left( \frac{r_0}{r} \right)^2 b_{\text{R}}(t) \tag{52}$$

$$B_{\text{T}} = B_{\text{T0}} \left( \frac{r_0}{r} \right) b_{\text{T}}(t) \tag{53}$$

$$B_{\text{N}} = B_{\text{N0}} \left( \frac{r_0}{r} \right)^{\kappa} b_{\text{R}}(t). \tag{54}$$

Here, $B_{\text{R0}}$, $B_{\text{T0}}$, and $B_{\text{N0}}$ are the mean magnetic field. $b_{\text{R}}$, $b_{\text{T}}$, and $b_{\text{N}}$ can be time-dependent such as the solar cycle (see section 2.2.3). The power-law index $\kappa$ is a free parameter and determines the dependence of $B_{\text{N}}$ on the inverse distance from the Sun $1/r$.

**2.2.3 Solar cycle dependence**

The solar cycle is a periodic change in the sunspot number over 11 years. In the plasma physics sense, the solar cycle is more associated with the magnetic activity of the Sun with a period of 22 years (the magnetic polarity is reversed after one sunspot cycle). During solar maximum the entire magnetic field of the Sun flips, thus alternating the polarity of the field every solar cycle. The solar (magnetic) activity is diverse such as solar radiation, ejections of solar material, and the number and the size of sunspots and the occurrence rate of solar eruptions. As a consequence, the periodic change in the solar magnetic field (or dipolar axis) affects the polarity of the IMF as well. To include the time dependent effect Kocifaj et al. (2006) suggests the following magnetic field model,

$$B_{\text{R}} = B_0 \left( \frac{r_0}{r} \right)^2 \cos \left( \frac{\pi t}{11[\text{yr}]} + \phi_0 \right) \tag{55}$$

$$B_{\text{T}} = -B_0 \left( \frac{r_0}{r} \right) \cos \vartheta \cos \left( \frac{\pi t}{11[\text{yr}]} + \phi_0 \right). \tag{56}$$

Here, $\vartheta$ is again latitude with $\theta = \pi - \vartheta$. Note that the transverse direction (with a unit vector $\boldsymbol{e}_{\text{T}}$ is constructed as $\boldsymbol{e}_{\text{T}} = \boldsymbol{\omega}_{\text{mag}} \times \boldsymbol{e}_{\text{R}}$, where $\boldsymbol{\omega}_{\text{mag}}$ is the magnetic axis of the Sun. If we assume that $\boldsymbol{\omega}_{\text{mag}}$ coincides with the rotation axis of the Sun, $\Omega_{\odot}$, then the relation $B_{\text{T}} = -B_{\phi}$ holds with $B_{\phi}$ given in Eqs. (17)–(18). However, in comparison with the second equation in Eqs. (17)–(18), the second equation in Eq. (35) differs by a factor $r_0 \Omega_{\odot}/U_r$ in addition to the inclusion of the time dependent terms. However, assuming solar wind speed $U_{\text{sw}} \simeq 450$ km s$^{-1}$, and solar rotation rate $\Omega_{\odot} \simeq 2\pi/24.47$ day$^{-1}$ this factor becomes close to unity at $r_0 = 1$ au.

**2.2.4 Polarity and tilt angle**

Two additional effects can further be incorporated into the IMF model, the polarity $A_{\text{mag}}$ and the tilt angle $\theta_{\text{tilt}}$. The polarity $A_{\text{mag}}$ is defined such that a case of $A_{\text{mag}} > 0$ corresponds to the magnetic fields pointing outward from the Sun in the northern hemisphere (the angle between the magnetic axis and the solar rotation axis is below 90°), and a case of $A_{\text{mag}} < 0$ is in the opposite sense to $A_{\text{mag}} > 0$. Using the polarity $A$, the Parker spiral magnetic field is given by the following equation

(Jokipii and Thomas, 1981):

$$\boldsymbol{B} = \frac{A_{\mathrm{mag}}}{r^2}\left(\boldsymbol{e}_r - \Gamma \boldsymbol{e}_\phi\right) \times$$
$$\left\{1 - 2H\left[\theta - \left(\frac{\pi}{2} + \theta_{\mathrm{tilt}}\sin\left(\phi - \frac{r\Omega_\odot}{U_{\mathrm{sw}}}\right)\right)\right]\right\}, \tag{57}$$

where $H$ is the Heaviside step function. $\Gamma$ is defined as

$$\Gamma = \frac{r\Omega_\odot \sin\theta}{U_{\mathrm{sw}}}. \tag{58}$$

The polarity $A_{\mathrm{mag}}$ is expressed in units of magnetic flux (cf. Eq. 23). An equivalent formulation of Eq. (57) is as follows (Kota and Jokippii, 1983):

$$\boldsymbol{B} = \frac{A_{\mathrm{mag}}}{r^2}\left(\boldsymbol{e}_r - \frac{r\Omega_\odot \sin\theta}{U_{\mathrm{sw}}}\boldsymbol{e}_\phi\right)\left[1 - 2H(\theta - \theta^*)\right] \tag{59}$$
$$\cot\theta^* = -\tan\theta_{\mathrm{tilt}}\sin\phi^* \tag{60}$$

where $\phi^*$ is the azimuthal angle in the co-rotating frame at an angular speed of the solar rotation,

$$\phi^* = \phi + \frac{r\Omega_\odot}{U_{\mathrm{sw}}}. \tag{61}$$

The tilt angle $\theta_{\mathrm{tilt}}$ is larger at near solar maximum and smaller at near solar minimum (Thomas and Smith, 1981), and typically varies from 75° at high level of solar activity to 10 down to 3° during solar minimum activity. A model of tilt angle variation over a 22-year solar cycle was constructed by Jokippii and Thomas (1981), Kota and Jokippii (1983) as follows:

$$\theta_{\mathrm{tilt}} = \theta_{\mathrm{t0}} + \theta_{\mathrm{t1}}\cos\left(\frac{2\pi t}{T}\right), \tag{62}$$

where $\theta_{\mathrm{t0}} = 20°$, $\theta_{\mathrm{t1}} = 10°$, and $T = 11\,\mathrm{yr}$. The tilt angle $\theta_{\mathrm{tilt}}$ is set to be at sunspot maximum at $t = 0$.

The wavy, flapping shape of the heliospheric current sheet is expressed by the equation for the polar angle as follows (Jokippii and Thomas, 1981):

$$\theta_{\mathrm{cs}} = \frac{\pi}{2} + \sin^{-1}\left[\sin\theta_{\mathrm{tilt}}\sin\left(\phi - \phi_0 + \frac{r\Omega_\odot}{U_{\mathrm{sw}}}\right)\right] \tag{63}$$
$$\simeq \frac{\pi}{2} + \theta_{\mathrm{tilt}}\sin\left(\phi - \phi_0 + \frac{r\Omega_\odot}{U_{\mathrm{sw}}}\right). \tag{64}$$

The approximation in Eq. (64) is valid for $\theta_{\mathrm{tilt}} \ll 1\,\mathrm{rad}$ (up to about 30°).

A sketch of the topology of the heliospheric current sheet is shown in Fig. 5, where the magnetic field is discontinuous, i.e. for vanishing $\theta - \theta^* = 0$ in $H(\theta - \theta^*)$. For small values of $\theta_{\mathrm{tilt}}$ the sheet is close to the plane defined in terms of the solar equator (left) while for larger values ($\theta_{\mathrm{tilt}} = 20°$) the wavy structure of the 'ballerina skirt' is found to be much more pronounced.

The drift motion depends on the sign of $qA_{\mathrm{mag}}$, a combination of the electric charge of the particle and the polarity of the solar magnetic field. During the period of $qA_{\mathrm{mag}} > 0$, the time variation of the cosmic ray flux shows a flatter maximum, while during $qA_{\mathrm{mag}} < 0$ the time variation of the cosmic ray flux shows a shape maximum, see, e.g. Jokippii and Thomas (1981) or Kota and Jokippii (1983).

[Figure]

**Fig. 5.** Shape of the 'ballerina skirt' model of the heliocentric current sheet defined by $\cos\theta^* = \cos\theta$. Topology at $t = 0$ and for $\theta_{t0} = 5^o$ (left) and $\theta_{t0} = 30^o$ (right).

A more refined magnetic field model is constructed by Burger et al. (2008), which offers an extension of the tilted heliospheric current sheet (with respect to the rotation axis) to the solar cycle dependence. The latitude-dependent magnetic field model is expressed as follows:

$$B_r = B_0 \left(\frac{r_0}{r}\right)^2 \tag{65}$$

$$B_\theta = B_r \frac{r}{U_{sw}} \omega^* \sin\beta^* \sin\phi^* \tag{66}$$

$$B_\phi = B_r \frac{r}{U_{sw}} \left[ \omega^* \sin\beta^* \cos\theta \cos\phi^* + \right.$$
$$\sin\theta (\omega^* \cos\beta^* - \Omega_\odot) + \frac{d\omega^*}{d\theta} \sin\beta^* \sin\theta \cos\phi^* +$$
$$\left. \omega^* \frac{d\beta^*}{d\theta} \cos\beta^* \sin\theta \cos\phi^* \right]. \tag{67}$$

Here

$$\phi^* = \phi - \Omega_\odot t + \frac{\Omega(r - r_0)}{U_{sw}} + \phi_0. \tag{68}$$

$B_0$ is again the radial component of the magnetic field at the reference radius $r_0$. The symbol $\beta_F$ is the angle (the Fisk angle) between the virtual magnetic axis (p-axis) and the rotation axis of the Sun, and $\omega$ is the differential rotation rate of the Sun. Both the angle $\beta_F$ and $\omega$ are generalized to the latitudinal dependent case by introducing the transition function $F_t(\theta)$ in the following way:

$$\beta^* = \beta_F F_t(\theta) \tag{69}$$

$$\omega^* = \omega F_t(\theta). \tag{70}$$

The transition function is constructed as follows (Burger et al., 2008):

$$F_t = |\tanh[\delta_{pol}\theta] + \tanh[\delta_{pol}(\theta - \pi)] - \tanh[\delta_{eq}(\theta - \theta_b')]|^2 \tag{71}$$

for the northern high-latitude region ($0 \leq \theta < \theta'_b$);

$$F_t = 0 \tag{72}$$

for the equatorial or low-latitude region ($\theta'_b \leq \theta \leq \pi - \theta'_b$); and

$$F_t = |\tanh[\delta_{pol}\theta] + \tanh[\delta_{pol}(\theta - \pi)] - \tanh[\delta_{eq}(\theta - \pi + \theta'_b)]|^2 \tag{73}$$

for the southern high-latitude region. $\theta'_b$ is the equatorward-limit polar angle of the coronal hole (characterized by open field lines) and is between $60°$ and $80°$ from the solar rotation axis in Burger et al. (2008). The symbols $\delta_{pol}$ and $\delta_{eq}$ are the control parameters of the transition from the high-latitude magnetic fields (Fisk-type model) into the low-latitude fields (Parker-type model), e.g., $\delta_{pol} = \delta_{eq} = 5.0$ proposed by Burger et al. (2008). The magnetic field model in Eqs. (65)–(67) represent a natural extension of the Parker model in that the case $F_t = 1$ reproduces the model proposed by Zurbuchen et al. (1997) and the case $F_t = 0$ the Parker model. The associated polar and azimuthal components of the flow velocity are:

$$U_\theta = r_0 \omega^* \sin\beta^* \sin\phi_\Omega \tag{74}$$

$$
U_\phi = r_0 \Bigg( \omega^* \sin\beta^* \cos\theta\cos\phi_\Omega + \omega^* \cos\beta^* \sin\theta +
$$
$$
\frac{d\omega}{d\theta} \sin\beta^* \sin\theta\cos\phi_\Omega +
$$
$$
\omega^* \frac{d\beta^*}{d\theta} \sin\theta\cos\phi_\Omega \Bigg). \tag{75}
$$

The Fisk angle $\beta_F$ is related to the tile angle of the heliospheric current sheet $\alpha_F$ by Burger et al. (2008):

$$\cos(\alpha_F + \beta_F) = 1 - (1 - \cos\theta'_{mm})\frac{\sin^2\alpha_F}{\sin^2\theta_{mm}}, \tag{76}$$

where $\theta_{mm}$ and $\theta'_{mm}$ are the equatorward (low-latitude) boundary of the polar coronal hole on the level of photosphere source surface in heliomagnetic coordinates, respectively. The boundary angles are expressed in heliographic coordinates as $\theta_b = \theta_{mm} - \alpha_F$ and $\theta'_b = \theta'_{mm} - \alpha_F$, respectively.

The tilt angles $\alpha_F$ and $\beta_F$ and the boundary angles $\theta_b$ and $\theta'_b$ can be modeled in a time-dependent way when constructing the Fisk-Parker-hybrid model (Burger et al., 2008) as a solar cycle dependent one: The time dependence of the tilt angle $\alpha_F$ is modeled as

$$\alpha_F = \alpha_{min} + \left(\frac{\pi}{4} - \frac{\alpha_{min}}{2}\right)\left[1 - \cos\left(\frac{\pi}{4}T[\mathrm{yr}]\right)\right] \tag{77}$$

for $0 \leq T[\mathrm{yr}] \leq 4\mathrm{yr}$, and

$$\alpha_F = \alpha_{min} + \left(\frac{\pi}{4} - \frac{\alpha_{min}}{2}\right)\left[1 - \cos\left(\frac{\pi}{7}(T[\mathrm{yr}] - 11)\right)\right] \tag{78}$$

for $4 < T \le 11\mathrm{yr}$, where $\alpha_{\mathrm{min}} = \pi/18$ is an offset tilt angle. Time $T$ is measured in units of years after a solar minimum. The time dependence of the boundary angles is

$$\theta_{\mathrm{b}} = \frac{\theta_{\mathrm{b(min)}}}{2} \left[ 1 + \cos\left(\frac{\pi}{4} T[\mathrm{yr}]\right) \right] \tag{79}$$

$$\theta_{\mathrm{b}}' = \frac{\theta_{\mathrm{b(min)}}'}{2} \left[ 1 + \cos\left(\frac{\pi}{4} T[\mathrm{yr}]\right) \right] \tag{80}$$

for $0 \le T \le 4\mathrm{yr}$, and

$$\theta_{\mathrm{b}} = \frac{\theta_{\mathrm{b(min)}}}{2} \left\{ 1 + \cos\left[\frac{\pi}{7} (T[\mathrm{yr}] - 11)\right] \right\} \tag{81}$$

$$\theta_{\mathrm{b}}' = \frac{\theta_{\mathrm{b(min)}}'}{2} \left\{ 1 + \cos\left[\frac{\pi}{7} (T[\mathrm{yr}] - 11)\right] \right\} \tag{82}$$

for $4 < T \le 11\mathrm{yr}$.

**3 Further models and effects**

**3.1 Magnetohydrodynamic models**

The models of the solar wind and the interplanetary magnetic field can be extended from kinematic or hydrodynamic treatments to magnetohydrodynamic (MHD) treatments. An overview of the MHD wind models is given by Tajima and Shibata (2002). Various magnetic effects are introduced in the MHD picture, e.g., the Alfvén velocity as a characteristic propagation speed (the Parker model, in contrast, recognizes the sound speed as a characteristic propagation speed) and the associated critical radius, collimation of the flow toward the rotation axis by magnetic pinching in the twisted field geometry.

**One-dimensional treatment**

An MHD model is proposed for an axi-symmetric, one-dimensional, centrifugal force driven wind on the solar equatorial plane (Weber and Davis, 1967). Six variables are determined as a function of the radial distance (mass density $\rho$, radial and azimuthal components of flow speed, $U_r$ and $U_\phi$, and that of the magnetic field, $B_r$ and $B_\phi$, and pressure $p$) using six equations (continuity equation, magnetic flux conservation, force balance, induction equation, adiabatic pressure, and energy conservation) and six integral constants (mass flux, magnetic flux, angular velocity of the Sun, Alfvén radius, entropy, and total energy). The Alfvén radius is defined as the radius at which the flow velocity reaches the Alfvén velocity in the radial component, $U_r = V_{\mathrm{A},r}$. At larger distances from the Sun, the solution is given asymptotically as

$$\rho \propto r^{-2} \tag{83}$$

$$U_r \to U_\infty \tag{84}$$

$$B_r \propto r^{-2} \tag{85}$$

$$B_\phi \propto r^{-1}. \tag{86}$$

The magnetic field becomes more azimuthal and thus twisted with increasing distance, $B_\phi/B_r \propto r$.

The momentum balance equation by Parker (1958) is extended to including the effect of magnetic field and Alfvén wave heating rate (Alazraki and Couturier, 1971; Belcher, 1971; Woolsey and Cranmer, 2014; Comişel et al., 2015):

$$\frac{1}{U}\frac{\mathrm{d}U}{\mathrm{d}r}\left(U^2 - U_\mathrm{c}^2\right) = -U_\mathrm{c}^2\frac{\mathrm{d}}{\mathrm{d}r}\ln B - c_\mathrm{s}^2\frac{\mathrm{d}}{\mathrm{d}r}\ln T + \frac{Q_\mathrm{A}}{2\rho(U+V_\mathrm{A})} - \frac{GM_\odot}{r^2}. \tag{87}$$

Here $Q_\mathrm{A}$ denotes the Alfvén wave heating rate. $U_\mathrm{c}$ is the critical speed

$$U_\mathrm{c}^2 = c_\mathrm{s}^2 + \frac{W_\mathrm{A}}{4\rho}\frac{3U+V_\mathrm{A}}{U+V_\mathrm{A}}, \tag{88}$$

where $W_\mathrm{A}$ is the energy density of the Alfvén waves including the perpendicular fluctuation components of the flow velocity $\delta U_\perp$ and that of the magnetic field $\delta B_\perp$,

$$W_\mathrm{A} = \frac{1}{2}\rho\delta U_\perp^2 + \frac{\delta B_\perp^2}{2\mu_0}. \tag{89}$$

**Two-dimensional treatment**

In the two-dimensional picture, the energy conservation (the generalized Bernoulli equation) and the conservation law perpendicular to the magnetic field (the generalized Grad-Shafranov equation) are derived using the force balance equation among the advection of the flow itself (flow nonlinearity such as steepening and eddies), the pressure gradient, the Lorentz force, and the gravitational attraction by the Sun, the mass flux conservation, the induction equation, and the adiabatic condition along the flow (Heinemann and Olbert, 1978; Sakurai, 1985; Lovelace et al., 1986). The generalized Grad-Shafranov equation cannot be solved analytically but needs to be solved numerically. It is found that the wind becomes collimated toward the rotation axis of the Sun (or the star) by the magnetic pinching of the spiral or twisted field. In fact, any stationary, axi-symmetric magnetized wind collimates toward the rotation axis at large distances (Heyvaerts and Norman, 1989).

It is useful to introduce the poloidal-toroidal expression of the magnetic field in the two-dimensional MHD treatment:

$$\boldsymbol{B} = \nabla \times (a\boldsymbol{e}_\phi) + B_\phi\boldsymbol{e}_\phi, \tag{90}$$

where $a$ denotes the magnetic stream function and $\boldsymbol{e}_\phi$ is the unit vector in the azimuthal direction around the rotation axis. The poloidal fields $B_\mathrm{p}$ (the first term in Eq. 90) are obtained by a family of curves under $a = \mathrm{const}$. We introduce the barred radius which is the distance from the rotation axis, $\bar{r} = r\sin\theta$. The flow velocity is decomposed by referring to the local magnetic field as

$$\boldsymbol{U} = \frac{\alpha_\mathrm{m}(a)}{\rho}\boldsymbol{B} + \bar{r}^2\Omega(a)\boldsymbol{e}_\phi, \tag{91}$$

where the first term (denoted by $U_\mathrm{p}$) is the flow velocity component parallel to the magnetic field in the frame rotating with the angular velocity $\Omega$, and the second term (denoted by $U_\phi$) is perpendicular to the magnetic field. The toroidal component of magnetic field is determined by the angular

momentum conservation,

$$\bar{r}\left(U_\phi - \frac{B_\phi}{\mu_0 a}\right) = l = \Omega \bar{r}_A^2(a), \tag{92}$$

where $l$ is the specific angular momentum and $\bar{r}_A$ is the Alfvén radius at which the poloidal component of the flow velocity becomes equal to the Alfvén speed for the poloidal component of the magnetic field. Equation (92) is obtained from the (steady-state) MHD momentum equation and the flow velocity expression in Eq. (91). The magnetic stream function needs to be determined for the flow velocity and the poloidal component of the magnetic field. The magnetic stream function is numerically evaluated from the momentum equation (or force balance) perpendicular to the magnetic field by solving the following equation (Sakurai, 1985):

$$\nabla \cdot \left[\left(\frac{\alpha_m^2}{\rho} - \frac{1}{\mu_0}\right)\frac{\nabla a}{\bar{r}^2}\right] = \rho\left(E' - \frac{1}{\gamma_p - 1}\frac{p}{\rho}\frac{K'}{K} + \bar{r}^2 \Omega \Omega'\right) +$$
$$\frac{B_p^2}{\rho}\alpha_m \alpha_m' +$$
$$D\left[\frac{D}{\mu_0}\Omega^2 \bar{r}^2 \alpha_m \alpha_m' - \alpha_m^2 \Omega^2 (\bar{r}_A^2)' - \alpha_m^2 \Omega \Omega' \left(\bar{r}_A^2 - \bar{r}_A\right)\right], \tag{93}$$

where

$$D = \frac{\mu_0 \rho \left(\bar{r}_A^2 - r^2\right)}{\bar{r}^2 (\mu_0 \rho \alpha_m^2 - \rho)} \tag{94}$$

and the prime $(\cdot)'$ denotes the differentiation with respect to the magnetic stream function, $d/da$. Equation (93) is the generalized Grad-Shafranov equation for the two-dimensional centrifugally-driven wind. The density $\rho$ follows the Bernoulli equation:

$$\frac{U_p^2}{2} + \frac{1}{2}(U_\phi - \Omega \bar{r})^2 + \frac{\gamma_p}{\gamma_p - 1}\frac{p}{\rho} - \frac{GM.}{r} - \frac{\Omega^2 \bar{r}^2}{2} = E(a) \tag{95}$$

under the polytropic or adiabatic equation of state

$$p = K(a)\rho^{\gamma_p}. \tag{96}$$

In the two-dimensional MHD treatment of the flow, the wind becomes collimated toward the rotation axis by the pinch of toroidal fields (Sakurai, 1985), causing a non-zero poleward (northward or southward) component of the magnetic field.

**3.2 More ingredients**

Solar wind models can further be improved by considering turbulent diffusion and pickup ions.

**Turbulent diffusion**

Turbulence on smaller spatial scales serves as an energy sink to large-scale mean fields, which leads to the notion of turbulent diffusion (mean-field electrodynamics). To see this more clearly, one may

decompose the magnetic field into a large-scale mean field $\boldsymbol{B}_0$ and a fluctuating field $\delta\boldsymbol{B}$ (with the zero mean value); and the flow velocity likewise:

405 $$\boldsymbol{B} = \boldsymbol{B}_0 + \delta\boldsymbol{B} \tag{97}$$

$$\boldsymbol{U} = \boldsymbol{U}_0 + \delta\boldsymbol{U}. \tag{98}$$

The induction equation for the large-scale magnetic field has then the frozen-in term for the large-scale fields $\boldsymbol{B}_0$ and $\boldsymbol{U}_0$ and the electromotive force term $\boldsymbol{\mathcal{E}}_{\mathrm{em}}$:

$$\frac{\partial \boldsymbol{B}_0}{\partial t} = \nabla \times (\boldsymbol{U}_0 \times \boldsymbol{B}_0) + \nabla \times \boldsymbol{\mathcal{E}}_{\mathrm{em}}. \tag{99}$$

The electromotive force is an averaged electric field coming from the coupling of the fluctuating with the fluctuating magnetic field by the cross product:

$$\boldsymbol{\mathcal{E}}_{\mathrm{em}} = \langle \delta\boldsymbol{U} \times \delta\boldsymbol{B} \rangle. \tag{100}$$

A widely-used model in the mean-field electrodynamics is that the electromotive force depends on the large-scale quantities such as the large-scale magnetic field, the curl of the large-scale magnetic field, and the curl of the large-scale flow velocity. By introducing the proper transport coefficients $\alpha_{\mathrm{t}}$, $\beta_{\mathrm{t}}$, and $\gamma_{\mathrm{t}}$, the electromotive force is modeled as

$$\boldsymbol{\mathcal{E}}_{\mathrm{model}} = \alpha_{\mathrm{t}} \boldsymbol{B}_0 - \beta_{\mathrm{t}} \nabla \times \boldsymbol{B}_0 + \gamma_{\mathrm{t}} \nabla \times \boldsymbol{U}_0. \tag{101}$$

After some algebra using Eqs. (99) and (101), one identifies that the term $\beta_{\mathrm{t}} \nabla \times \boldsymbol{B}_0$ becomes nothing other than the diffusion term for the large-scale magnetic field (under the condition that the coefficient $\beta_{\mathrm{t}}$ is not negative):

$$\frac{\partial \boldsymbol{B}_0}{\partial t} = \nabla \times (\boldsymbol{U}_0 \times \boldsymbol{B}_0) + \nabla \times (\alpha_{\mathrm{t}} \boldsymbol{B}_0) + \beta_{\mathrm{t}} \nabla^2 \boldsymbol{B}_0 + \nabla \times (\gamma_{\mathrm{t}} \nabla \times \boldsymbol{U}_0). \tag{102}$$

The terms with $\alpha_{\mathrm{t}}$ and $\gamma_{\mathrm{t}}$ in turn may amplify the large-scale magnetic field when the coefficients are in favor of field amplification (dynamo mechanism). The transport coefficients are theoretically estimated as follows:

410 $$\alpha_{\mathrm{t}} = C_\alpha \tau (-h_{\mathrm{kin}} + h_{\mathrm{cur}}) \tag{103}$$

$$\beta_{\mathrm{t}} = C_\beta \tau (e_{\mathrm{kin}} + e_{\mathrm{mag}}) \tag{104}$$

$$\gamma_{\mathrm{t}} = C_\gamma \tau h_{\mathrm{crs}}, \tag{105}$$

where $C_\alpha$, $C_\beta$, and $C_\gamma$ are dimensionless scalar factors, and are estimated as (Yoshizawa, 1998),

$$C_\alpha \simeq 0.02 \tag{106}$$

415 $$C_\beta \simeq 0.05 \tag{107}$$

$$C_\gamma \simeq 0.04. \tag{108}$$

The symbol $\tau$ denotes the turbulent correlation time length, and $h$ and $e$ represent the helicity and the energy quantities: $h_{\mathrm{kin}}$ the kinetic helicity density, $h_{\mathrm{cur}}$ the current helicity density, $h_{\mathrm{crs}}$ the cross helicity density, $e_{\mathrm{kin}}$ the turbulent kinetic energy density, and $e_{\mathrm{mag}}$ the turbulent magnetic energy density. The helicity density quantities and the energy density quantities are defined for the fluctuating field,

$$h_{\mathrm{kin}} = \langle \delta \boldsymbol{U} \cdot (\nabla \times \delta \boldsymbol{U}) \rangle \tag{109}$$

$$h_{\mathrm{cur}} = \frac{1}{\mu_0 \rho_0} \langle \delta \boldsymbol{B} \cdot (\nabla \times \delta \boldsymbol{B}) \rangle \tag{110}$$

$$h_{\mathrm{crs}} = \frac{1}{\sqrt{\mu_0 \rho_0}} \langle \delta \boldsymbol{U} \cdot \delta \boldsymbol{B} \rangle \tag{111}$$

$$e_{\mathrm{kin}} = \frac{1}{2} \langle |\delta \boldsymbol{U}|^2 \rangle \tag{112}$$

$$e_{\mathrm{mag}} = \frac{1}{2\mu_0 \rho_0} \langle |\delta \boldsymbol{B}|^2 \rangle. \tag{113}$$

Note that different definitions are possible for the helicity and energy density quantities. In the definition above (Eqs. 109–113) the fluctuating magnetic field is converted into the velocity dimension such as $\delta \boldsymbol{B} / \sqrt{\mu_0 \rho_0}$ and the energy density is represented as that per unit mass. The correlation time length $\tau$ can in the simplest case be modeled or represented by the eddy turnover time,

$$\tau_{\mathrm{ed}} = \frac{\ell}{\delta U} = \frac{e_{\mathrm{kin}} + e_{\mathrm{mag}}}{\varepsilon}, \tag{114}$$

where $\varepsilon$ is the dissipation rate which needs to be obtained by solving an equation in the similar fashion to the turbulence energy (Yokoi et al., 2008). The estimate of time scale can be extended by including the Alfvén time effect into a synthesized time scale $\tau_{\mathrm{s}}$ in the additive sense in the frequency domain as

$$\frac{1}{\tau_{\mathrm{s}}} = \frac{1}{\tau_{\mathrm{ed}}} + \chi \frac{1}{\tau_{\mathrm{A}}}, \tag{115}$$

where $\tau_{\mathrm{A}}$ denotes the Alfvén time

$$\tau_{\mathrm{A}} = \frac{\ell}{V_{\mathrm{A}}} = \frac{|e_{\mathrm{kin}} + e_{\mathrm{mag}}|^2}{\varepsilon V_{\mathrm{A}}^2} \tag{116}$$

with the length scale $\ell$ and the Alfvén speed $V_{\mathrm{A}}$. The symbol $\chi$ is the weight factor for the Alfvén time, and is estimated to be of the order $10^2$ in the solar wind application (Yokoi et al., 2008). A more rigorous treatment is to solve two sets of equations, one for the large-scale mean fields and the other for the small-scale turbulent fields. This task can be achieved either analytically using the two-scale direct interaction approximation (Yokoi, 2006; Yokoi and Hamba, 2007; Yokoi et al., 2008) or numerically (Usmanov et al., 2012, 2014, 2016).

**Pickup ions**

Pickup ions from interstellar neutral hydrogen atoms are one of the ingredients to the solar wind, and contribute to additional mass of the plasma, which results in deceleration of the solar wind expansion and in increase in the plasma temperature. Pickup ions originate in (1) charge exchange with

the solar wind protons and (2) photoionization by the solar radiation. Steady-state MHD equations for the wind including pickup ions are introduced by Isenberg (1986) and Whang (1998), and are numerically implemented to simulation studies for a three-component fluid (thermal protons, electrons, pickup protons) by Usmanov and Goldstein (2006); Usmanov et al. (2014) and for a four-component fluid by adding interstellar hydrogen (Usmanov et al., 2016).

The continuity equation in the one-fluid sense (mixture of electrons, solar wind protons, and pickup ions of interstellar origin) has a contribution from the photoionization as a source term. and is written for the steady state as (Whang, 1998)

$$\nabla \cdot (\rho \boldsymbol{U}) = m_{\mathrm{p}} q_{\mathrm{ph}}, \tag{117}$$

where $\rho$ and $\boldsymbol{U}$ denote the mass density and the flow velocity in the one-fluid sense, $m_{\mathrm{p}}$ the proton mass, and $q_{\mathrm{ph}}$ the pickup ion production rate by the photoionization process,

$$q_{\mathrm{ph}} = \nu_0 \left( \frac{r_0^2}{r} \right) n_{\mathrm{nt}}. \tag{118}$$

Here $\nu_0 = 0.9 \times 10^{-7}$ s$^{-1}$ is the photoionization rate per hydrogen atom at the Earth orbit distance as reference $r_0 = 1$ au, and $n_{\mathrm{nt}}$ is the number density of neutral hydrogen (of interstellar origin). The one-fluid momentum equation in the steady state is approximated into (by neglecting higher-order terms) (Whang, 1998)

$$\rho \boldsymbol{U} \cdot \boldsymbol{U} + \nabla P - \rho \nabla \left( \frac{GM_\odot}{r} \right) - \frac{1}{\mu_0} (\nabla \times \boldsymbol{B}) \times \boldsymbol{B} = -(q_{\mathrm{ex}} + q_{\mathrm{ph}}) m_{\mathrm{p}} \boldsymbol{U}. \tag{119}$$

Here $q_{\mathrm{ex}}$ is the pickup ion production rate by the charge exchange process,

$$q_{\mathrm{ex}} = \sigma_{\mathrm{ex}} n_{\mathrm{sw}} n_{\mathrm{nt}} U, \tag{120}$$

where $\sigma_{\mathrm{ex}}$ is the cross section of charge exchange between a hydrogen atom and the solar wind protons, $n_{\mathrm{sw}}$ is the number density of solar wind protons.

**3.3 Stellar wind and interstellar space**

Various outflow models have been proposed for the stellar wind. For example, a wind model is constructed and numerically studied for the thermally-driven hydrodynamic outflow from low-mass stars (Johnstone et al., 2015). A dead zone due to the magnetic dipole field effect can arise in the equatorial region (Keppens and Goedlbloed, 1999). A model is also constructed for the stellar winds around asymptotic giant branch (AGB) stars with dust grains by employing the MHD equation for the stellar wind plasma and the Euler equation for the dust grains under the gravity, the radiation pressure, and the drag force (Thirumalai and Heyl, 2010), showing the possibility of a stellar wind driven by dust grains. Mass-loss rate is observationally studied via stellar winds for sub luminous stars (Krtička et al., 2016), in which the following flow velocity model is used for fitting with three

parameters $U_1$, $U_2$, and $\gamma_{sw}$:

$$U = \left[ U_1 \left( 1 - \frac{R_s}{r} \right) + U_2 \left( 1 - \frac{R_s}{r} \right)^2 \right] \left\{ 1 - \exp \left[ \gamma_{sw} \left( 1 - \frac{r}{R_s} \right)^2 \right] \right\}, \tag{121}$$

445   where $R_s$ is the stellar radius.

Stellar winds can be detected by the spectroscopic investigation. A line spectrum becomes distorted to blue-shifted absorption and redshifted emission by the retarding stellar wind (away from the observer), known as the P Cygni profile. One type of the stellar wind models is the Lucy model (Lucy, 1971):

$$U = U_t \left[ 1 - \frac{(1 - a_{sw}) R_s}{r} - a_{sw} \frac{R_s^2}{r^2} \right]^{1/2}, \tag{122}$$

where $a_{sw}$ is a free parameter with $-1 < a_{sw} < 1$. Equation (122) satisfies the conditions of zero speed at the stellar surface, ($U = 0$ at $r = R_s$) and asymptotic behavior at very large distances from the star $U \to U_t$ as $r \to \infty$). $U_t$ is the terminal flow velocity. The flow speed increases monotonously as a function of the radius, $U > 0$ and $\frac{dU}{dr} > 0$. The other type is a variant of the Lucy model (Kudritzki and Puls, 2000):

$$\frac{U}{U_t} = \left( 1 - b_{sw} \frac{R_s}{r} \right)^{\beta_{sw}}, \tag{123}$$

where the constant $b_{sw}$ is the flow velocity at the inner boundary of the stellar wind. An even more simplified expression is (Lamers, 1998)

$$\frac{U}{U_t} = \left( 1 - \frac{R_s}{r} \right)^{\beta_{sw}}, \tag{124}$$

where $U_t$ is the asymptotic, termination flow speed. $\beta_{sw}$ is a free parameter, and is empirically chosen as $0.5 \leq \beta_{sw} \leq 4$ (Sapar et al., 2003).

**4   Summary and conclusions**

There is an increasing amount of models for the interplanetary magnetic field. Starting with the
450   Parker model, the magnetic field model can be extended to include the latitudinal dependence, the poleward component, the time-dependence, and the polarity and tilt effect even in the analytic or semi-analytic treatment. Which model to choose would depend on the application, e.g., if the solar cycle is to be included or not, or if the latitudinal dependence is to be or not. In the temporal sense, cosmic ray diffusion has the shortest time scale, about 13 hours for relativistic particles nearly at
455   the speed of light to travel over 100-au distance in the heliosphere. In contrast, plasma turbulence evolves together with the solar wind, and the time scale is intermediate, being of the order or days, cf. the solar wind travel time from the Sun to the Earth orbit, 1 au, is about 100 hours or roughly 4 days. Charged dust motions and modulation of the cosmic ray flux in the heliosphere evolve on the longest time scale among the three applications, of the order of of years (secular variation of the
460   orbital parameters).

The accuracy or the uncertainty of the reviewed models need to be verified using in situ magnetic field measurements from the previous, current, and upcoming spacecraft missions. Above all, the magnetic field in the inner heliosphere will be extensively studied with Parker Solar Probe, Bepi-Colombo (in particular, the cruise-phase measurements), and Solar Orbiter.

465    It is interesting to note that the analytic expression is also available for the coronal magnetic field (during the solar minimum) and the local interstellar magnetic field surrounding the heliosphere. Hence, naively speaking, one may expect to construct a more complete model of the magnetic field from the Sun to the local interstellar medium. Such a model, once smoothly and rationally connected from one region to another, enables one to improve the accuracy of theoretical studies on plasma

470    turbulence evolution, charged dust motions, and diffusion of cosmic ray and energetic particles.

It is also worth noting the limits of the models. First, the magnetic fields are highly structures in the solar corona and at the solar surface. At some distance sufficiently close to the Sun, the interplanetary magnetic field should smoothly be connected to the coronal magnetic field. Second, the outer heliosphere has the termination shock and the heliopause, which are not included in the

475    models in this review. Third, the solar variability includes not only the 11-year sunspot number variation or the 22-year magnetic structure variation, but also modulations of the solar cycle on long time scales such as 100 or even 1000 years.

*Acknowledgements.* This work is financially supported by Austrian Space Applications Programme FFG ASAP-12 SOPHIE at Austrian Research Promotion Agency under contract 853994 and Austrian Science Funds (FWF)

480    under contract P30542-N27.